# Understanding Drainage Dynamics and Irrigation Management in a Semi-Arid Mediterranean Basin

Víctor Altés [1,*], Joaquim Bellvert [2], Miquel Pascual [3] and Josep Maria Villar [1]

1   Department of Environment and Soil Sciences, Universitat de Lleida, 25198 Lleida, Spain
2   Efficient Use of Water in Agriculture Program, Institute of AgriFood Research and Technology (IRTA), Fruitcentre, Parc Científic i Tecnològic Agroalimentari de Gardeny (PCiTAL), 25003 Lleida, Spain
3   Department of Horticulture, Botany and Gardening, Universitat de Lleida, 25198 Lleida, Spain
*   Correspondence: victor.altes@udl.cat

**Abstract:** Irrigation is one of the main users of water worldwide and its overuse may affect the natural regimes of water systems. To avoid this, drainage and irrigation management needs to be improved. This study aims to determine the amount of water lost to drainage in a semi-arid Mediterranean irrigated area. Water use, rainfall and drainage were monitored for 12 months (2019–2020) in a 425 ha sub-basin in the Algerri-Balaguer irrigation district (8000 ha, NE Spain). In addition, irrigation requirements were estimated using the single-crop FAO-56 method and a two-source energy balance model (TSEB) was used to estimate actual evapotranspiration in the sub-basin. Water lost to drainage in the sub-basin was estimated as 18% of the total water that entered the perimeter as irrigation and rainfall, which are almost five time higher than theoretical requirements of leaching. Out of the total drainage water, 57% was estimated to be irrigation water and 43% rainwater. The average amount of irrigation water used was 614 mm and irrigation efficiency in the sub-basin was estimated at 80.2% and averaged actual evapotranspiration at 1144 mm. The available margin of improvement is between 19.3% of the present irrigation drainage ratio and the 3.8% estimated with the leaching requirement model.

**Keywords:** water use; irrigation management; drainage; evapotranspiration; remote sensing; water balance

## 1. Introduction

Worldwide water demand is rapidly increasing, and is expected to have risen by 55% by 2050 [1]. In this scenario, improving the efficiency of irrigation water management must be a priority. Given that irrigation consumes more than 90% of the water used in agricultural areas of low population and scarce industrial activity, it can confidently be stated that agriculture causes a significant pressure on water resources [2]. As an added complication, competition for water is normal, especially among users, including industrial activities, that depend on its availability for economic returns.

When irrigation is implemented, the soil–plant–atmosphere continuum (SPAC) [3] has to be considered. This means that factors such as soil type, rainfall, evapotranspiration [4] and crop type play an essential role in the goal of achieving properly managed irrigation. As well as many other aspects, irrigation management can involve the construction of canals and/or ponds, land leveling, irrigation system design and irrigation scheduling [5].

One of the main management issues than needs to be dealt with at farm and basin scale is drainage, which can be defined as the removal of excess surface and subsurface water from the land, which includes the removal of soluble salts from the soil [6] to enhance crop growth, maintain soil quality and avoid waterlogging.

Water lost from the soil as drainage may cause downstream impacts [7–10], as it can be loaded with salts and other pollutants. Therefore, it is necessary to reduce drainage produced during irrigation to the minimum that is required. In order to do so, Doorenbos

and Pruitt [11] defined the leaching requirement (LR) as the portion of irrigation water applied that must drain through to the active root zone to remove accumulated salts, which at the same time will depend on the quality of the irrigation water. This quality can be represented as the irrigation water electrical conductivity ($EC_w$), as it expresses the property of the water to transfer an electrical charge, or the salt content of the water.

Any amount of irrigation water above the LR can be considered as non-used water by the plant, expressed as the leaching fraction (LF) [12], which is the real water that went through the root zone. The difference between the LR and LF can be defined as the drainage surplus. It is also necessary to consider rainfall when talking about the LF, as it is one of the main contributors to the water balance and to soil drainage.

If the impact of irrigation downstream is to be minimized, the drainage surplus should be as small as possible. Drainage also needs to be considered if the aim is to achieve high irrigation efficiencies, and when adjusting water demands and estimating water balances. The importance of drainage is therefore evident, as is the need for drainage monitoring and modeling wherever possible.

For many years, research studies related to water management have focused on irrigation requirements, water balances, irrigation efficiencies and water productivity, among others. However, few of them have addressed the topic of agricultural drainage. A quick search in Scopus, comparing published scientific articles on irrigation (176,659) with articles which, in the title, abstract or key words, include agricultural drainage and irrigation (13,069), revealed that just 7.4% of the articles on irrigation included the word 'drainage'. This may be attributable to the difficulties of its monitoring and its being an issue that extends beyond the field or district level.

Knowing the dynamics of drainage at the watershed level, as well as the quantities and quality of unconsumed irrigation water, facilitates evaluations of its possible reuse and quantifications of its impact. Agricultural drainage studies are a necessary contribution to understand the impacts on natural systems and to realistically quantify the effect of irrigation on the environment [10]. In addition, the contribution of non-agricultural sources such as urban wastewater treatment plants must also be considered.

To analyze and improve irrigation management in semi-arid areas, the methods that have been used include water balances [13–15], irrigation performance studies [16,17] and nitrate and salt balances [18–20].

One of the tools that is being used largely for studying the environment and landscape is remote sensing, which is the collection and interpretation of information about an object, area or event without being in physical contact with it [21]. The term "remote sensing" was first coined in the early 1960s to describe any means of observing the Earth from afar, particularly as applied to aerial photography, the main sensor used at that time [22]. In this regard, remote sensing technologies have been widely used to account for water balances at regional [23,24] and plot scales [25]. It is also possible to estimate evapotranspiration by remote sensing, either through empirical [26] or energy balance [27–29] models. Remote sensing has additionally been used to estimate soil moisture at different scales [30,31] or to obtain more accurate irrigation-related parameters such as crop coefficients ($K_c$) [32] and crop evapotranspiration [33].

The main objective of this study was to quantify the amount of irrigation water lost through drainage in a semi-arid Mediterranean irrigation district. In addition, the secondary objectives included: (i) determination of the fraction of rainwater in the drainage water, (ii) estimation of the water balance variables at sub-basin scale using in situ data and (iii) a comparison, at sub-basin scale, of the actual evapotranspiration obtained through a water balance ($ET_{a\,B}$) approach with that estimated through remote sensing using a surface energy balance model ($ET_{a\,TSEB}$).

## 2. Materials and Methods

### 2.1. Study Area Description

The study was conducted in the Algerri-Balaguer (AB) irrigation district (NE Spain) (Figure 1). The area was dryland until 1998. The current total area occupies 8000 ha, but the extension equipped to supply water is 6858 ha (2021). In order to implement irrigation, actions on land consolidation, the installation of a pressurized irrigation network and a drainage network and the establishment of Special Protection Areas (SPAs) were developed from 1998 to 2017 by the public regional infrastructure company (Infraestructures.cat). The drainage network was built following the natural drainage system existing in the zone previously. Prior studies in irrigation and drainage were carried out during 2006 and 2008, when lower amount of land was irrigated [34].

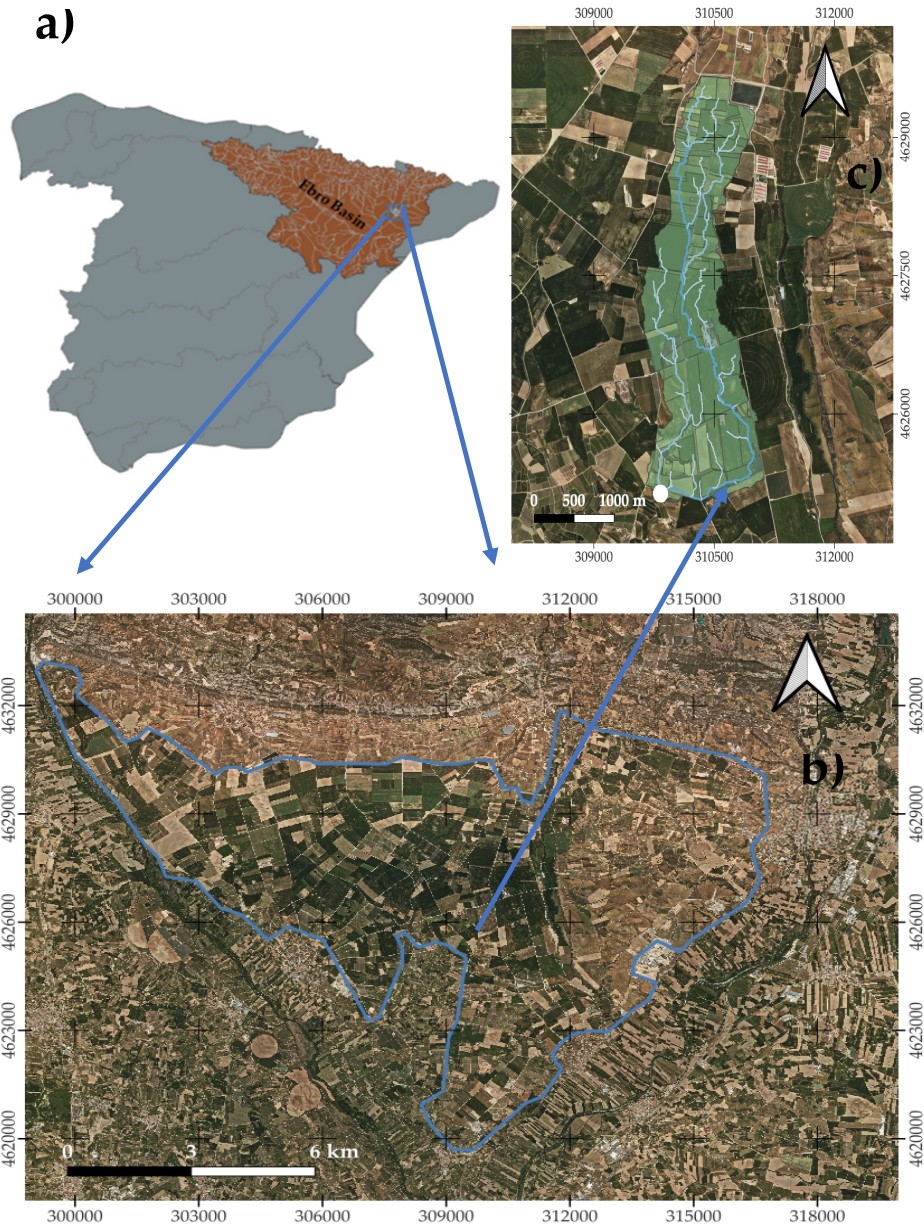

**Figure 1.** Study site showing in (**a**) the location of the study area in the Ebro Basin, (**b**) the Algerri-Balaguer irrigation district and (**c**) the AB5 sub-basin, drainage network, farm distribution and outlet position (white dot). Maps developed according to the Coordinate Reference System ETRS89/UTM zone 31N and Authority ID EPSG:25831.

From a geological point of view, the study area is located in the piedmont zone between the Pre-Pyrenean Mountain system, formed by large mountain ranges and the Tertiary depression, dominated by the Ebro Valley. The area is delimited to the north by Serra Llarga, which constitutes the southern flank of the Barbastro-Balaguer Anticline. This is connected by the Alfarràs plateau covered by fluvial deposits and glacis, acting as a link with the most depressed areas. To the south of the study area, the limit is marked by the central-eastern depression, crossed by the Segre and Noguera Ribagorçana Rivers that develop important stepped terraces.

The stratigraphy of this area includes Tertiary materials formed from base to top by gray gypsum and two detrital units. The lower unit, formed by stratified limestones, marls and flint intercalations, sandstones alternating with silts and clays and carbonate levels, to which is superimposed the upper detrital unit composed of paleochannel deposits formed by sandstones, silts and red clays, alternating with sandstone bars and conglomerates. Above, Quaternary deposits have been described, in which terraces, colluvial deposits and glaciers have been differentiated, the latter being the best represented within the study area [35].

The area is located in the hydrological basins of the Segre and Noguera Ribagorçana Rivers. Regarding the hydrogeological framework, it is worth mentioning the aquifer system of the alluvial terraces of the Ebro, subdivided into two aquifers, the lower alluvial terrace of the Noguera Ribagorçana and its current floodplain, and the last alluvial terrace of the Segre River. In both cases the impermeable substrate is formed by tertiary lithologies (clays, gypsum and loams), and the recharge is produced by direct infiltration of the rivers, infiltration of rainwater, scarce infiltration of lateral torrents and fundamentally by infiltration of irrigation returns [36].

More precisely, the studied area (Figure 1c) is located on glacis deposits of different generations correlated with different terraces of the Noguera Ribagorçana and Segre Rivers. In this case, the oldest ones attributed to first- and second-generation glacis correlate with terraces of heights 55–60 m and 25–30 m, respectively. The most modern, fourth generation glacis are associated with the lower part of the current fluvial courses, valley bottoms and alluvial deposits. These deposits are formed by extensive detrital materials that create extensive plains whose lithology of pebbles, gravels, sands and silts differs from the terraces, due to their poor arrangement, their high variability in size and their high content of clays and gypsiferous silts, which result in practically impermeable materials. Further details from the study area can be found online in Mapa Geológico de España. E.1:50.000-Hoja 359-BALAGUER-Conjunto de datos|datos.gob.es

Concerning the soils in the Algerri-Balaguer irrigation district, a survey of soil resources carried out during 1991 [37] resulted in a detailed soil map (1:25,000), which is digitally available [38]. In general, soils are calcareous, deep and of medium texture, with the presence of gravels only in the terraces of the rivers that surround the area. Soils in the study area (Figure 1c) are variable, but mainly characterized by having good drainage, with exceptions in some occasional spots near the natural drainage system in the lowest areas. These soils have a pH between 7.8 and 8.4, with a low organic matter level (1–2%) and low cation exchange capacity (6–12 $cmol_c$/kg) with a medium-high content of calcium carbonate equivalent (10–50%). The electrical conductivity of saturated paste extract ($EC_e$) (1:5) is between 0.1 and 2 dS/m in most soils. The use of manure, slurry and no-till farming in the last 20 years has contributed to soil fertility, improving the levels of organic matter in the upper horizons. Moreover, data related to soil hydrology from the 5 main types of soils in the studied sub-basin are shown in Table 1. It is important to highlight that the main subjacent material in the study zone consists in impervious layers as lutite or cemented gravels which minimizes water seepage and capillary rise in the area.

**Table 1.** Type of soil [38], USDA Soil Taxonomy [39], depth of the soil, hydraulic conductivity [40], infiltration velocity [40] and subjacent material of main soils in the studied area according to the "Detailed Soil Survey Map of Catalonia of the Irrigated Area by Algerri-Balaguer Canal" [37].

| Type of Soil Series | USDA Soil Taxonomy | Depth (m) | Hydraulic Conductivity (m/Day) | Infiltration Velocity (m/Day) | Subjacent Material |
|---|---|---|---|---|---|
| Bellcaire | Typic Calcixerepts | 0.8–1.2 | 0.12–1.6 | 0.12–1.6 | Lutite |
| Bellvís | Petrocalcic Calcixerepts | 0.4–0.8 | 1.6–3.1 | 0.5–3.1 | Lutite |
| Comelles | Typic Xerofluvents | >1.20 | 0.5–1.6 | 0.1–0.5 | Lutite |
| Cava | Xeric Petrocalcids | 0.2–0.4 | 3.1–6 | 3.1–6 | Cemented Gravel |
| Seana | Xeric Petrocalcids | 0.4–0.6 | 3.1–6 | 3.1–6 | Cemented Gravel |

In the AB district, the main land use is double cropping, which consists of planting more than one crop per year. Maize (*Zea mays* L.) is usually the main crop. This staple crop is planted in April, but in the case of double cropping is planted immediately after the harvest of the first crop, mainly barley (*Hordeum vulgare* L.), wheat (*Triticum aestivum* L.) or pea (*Pisum sativum* L.). This occurs during the second half of June. Forage crops such as alfalfa (*Medicago sativa* L.) and ray grass (*Lolium sp.* L.) are also common in the area. Less common crops are onion (*Allium cepa* L.) and colza (*Brassica* sp.). A significant area of land is left to fallow in accordance with EU requirements. Occasionally, some minority crops are also grown in the area, such as potatoes (*Solanum tuberosum* L.), soybeans (*Glicine max* L.) Merr, sunflowers (*Helianthus annuus* L.) or sod crops. The permanent woody crops are, in order of importance, peach (*Prunus persica* (L.) Batsch.), pear (*Pyrus communis* L.), olive (*Olea europaea* L.), almond (*Prunus dulcis* Mil.), apple (*Malus domestica* Borkh.), grapevine (*Vitis vinifera* L.), apricot (*Prunus armeniaca* L.) and walnuts (*Julgans regia* L.). These crops are present in 1390 ha (2021), which amounts to 20% of the total irrigated area.

Water used in the irrigation district is pumped 32 m from the Noguera Ribagorçana River a highly regulated river with four dams, belonging to the Ebro River basin, north of the city of Lleida (NE Spain), and south of the Barbastro-Balaguer anticline (Figure 1). Water demanded by the irrigation district, with a maximum flow of 4.8 m$^3$/s, is released from the Santa Ana dam at any time of the year, a water reservoir of 236 hm$^3$ located 380 m above sea level (m.a.s.l.) (UTM X: 797,086.1, UTM Y: 4,642,949.4), in addition to the ecological flow of 6.8 m$^3$/s (on average) from the river. A 15-km channel delivers the pumped water to two main reservoirs. From these reservoirs, water is pumped again to 4 reservoirs located at a higher elevation, allowing pressurized water (4.5 kg/cm$^2$) to reach all the hydrants in the irrigation area. For this reason, the main concern of the irrigation district is the energy cost. The quality of the irrigation water is almost constant during all the year, with electrical conductivity (EC$_w$) between 0.3 and 0.4 dS/m (averaged on 0.37 dS/m). The ranges of all the parameters are normal [41], and the nitrate content is usually low for agricultural crops (less than 0.05 meq NO$_3^-$/L).

All the irrigation systems in the area are pressurized, with sprinkler and drip irrigation systems used for herbaceous and woody crops, respectively. Water is supplied to the farm fields by delivery points (hydrants). The total number of hydrants is 1351, making an average of 5.07 ha/hydrant. These delivery points also facilitate the supply of water to pig, cattle and poultry farms.

The main drainage network is designed to evacuate excess irrigation and rainfall water and avoid a rise in the water table, found at depths from 0.5 to 2.15 m in some occasional spots near the natural drainage system in the lowest areas during the soil survey study conducted in the area in 1991 [37]. The artificial drainage network consists primarily of a combination of open drains (surface ditches) with corrugated plastic pipes between 200 and 400 mm in diameter, covered by geotextile and gravels to avoid pipe clogging. In some

end sections, concrete pipes of 600 to 1200 mm diameter are used. The surface ditches are usually 2 m deep, 1 m wide at the base and around 2.5 m wide at the surface. In some cases, farmers install corrugated plastic drains in areas with waterlogging problems and connect them to the main drainage network. These modern facilities enable monitoring and control of irrigation, rainfall and drainage water in the area.

This region is characterized by a dry continental Mediterranean climate, with an average air temperature ($T_{air}$) of 14.4 °C, cold winters and dry and hot summers. Average annual rainfall (R) and reference evapotranspiration ($ET_o$) are 378 mm and 1072 mm (2000–2021), respectively [4]. This semiarid land has an aridity index of 0.35 (380/1072) [42]. Seasonal rainfalls are common during autumn and spring. Meteorological data were obtained from the Albesa agrometeorological station (UTM X: 306,325.0, UTM Y: 4,625,793.0) at 267 m.a.s.l., part of the XEMA regional network [43] at daily basis during the period 2000–2021.

The area is divided into several sub-basins. The study was developed in the AB5 minor sub-basin (425 ha), from 15 October 2019 to 15 October 2020. During that period, R and $ET_o$ were 533 and 1045 mm, respectively. Average air temperature was 14.9 °C (Figure 2).

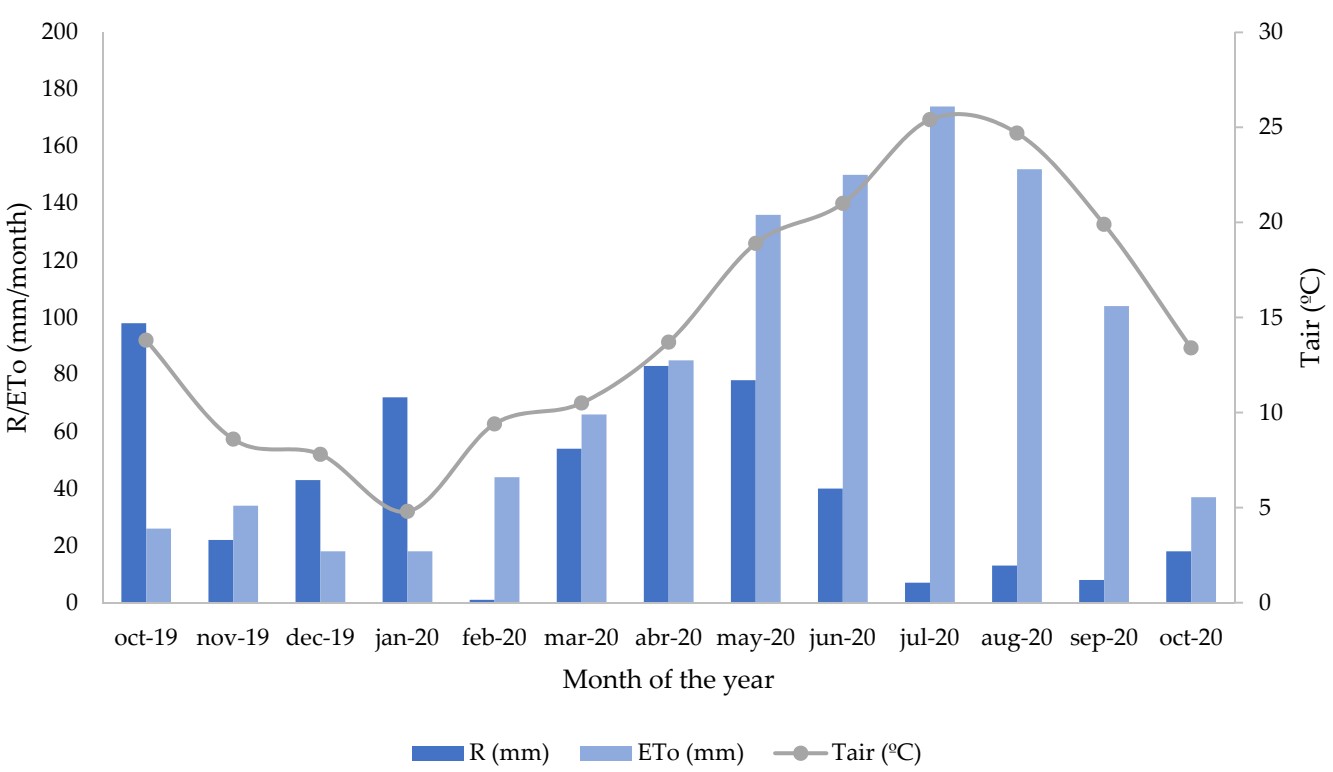

**Figure 2.** Average daily temperature (Tair), monthly cumulative rainfall (R) and monthly reference evapotranspiration ($ET_o$) during the study period in the Albesa station. From 15 October 2019 to 15 October 2020. Data obtained from the meteorological regional network [43].

The sub-basin was delineated in QGIS 3.10 software, with the 'SAGA Terrain Analysis: Channel network and drainage basins' toolbox. Prior to that, a 'Fill sinks' [44] algorithm was applied in a digital elevation model (DEM) of the study zone downloaded from the ICGC [45] (Figure 1c).

*2.2. Water Balance*

2.2.1. Water Balance Equation

A water balance approach was applied at sub-basin scale to calculate the actual evapotranspiration (ET$_{a\,B}$) considering Equation (1), which consists of assessing the inputs and outputs of water into the root zone during the study period [4]:

$$ET_{a\,B} = I + R - RO - DP + CR \pm \Delta SF \pm \Delta SW \tag{1}$$

where ET$_{a\,B}$ is the actual evapotranspiration, I is the total irrigation water applied in the sub-basin, R is the total rainfall in the sub-basin and $\Delta$SW is the change in soil water content, which was assumed as 0 during the study period, as it is developed during an entire hydrological year. Also taken into consideration is the runoff during rain episodes that reaches the outlet (RO), as well as deep percolation (DP), of rainfall and irrigation origin, which we consider as drainage water (D). Capillary rise (CR) and subsurface flow (SF) were discarded, as these two variables are difficult to assess and can be ignored in short periods [4]. Consequently, the equation for estimating evapotranspiration is simplified as follows:

$$ET_{a\,B} = I + R - RO - DP \tag{2}$$

2.2.2. Irrigation and Crop Water Requirements

The AB5 sub-basin has 40 hydrants which supply irrigation water to 419 ha (on average 10.5 ha/hydrant). In the case of this sub-basin, there are no livestock facilities that are supplied with water. Irrigation is carried out mainly at night to avoid soil water evaporation in the hottest hours and wind drift losses. Usually, irrigation seasons correspond to the period from mid-March to mid-October. The total volume of irrigation water in the AB5 basin during the 2019–2020 growing season was measured using a water counter (CZ4000 water meter, Contazara, Spain) installed in each of the hydrants. Data were recorded at the beginning and at the end of the study period.

During the study period, different crops were grown in the area. Ninety-eight percent of the sub-basin was covered by crops, and crop type classification was obtained from DUN-SIGPAC 2020 [46] (Table 2). The Geographic Information System for Agricultural Parcels (SIGPAC) allows the geographic identification of parcels declared by farmers and livestock farmers for any subsidy scheme related to the area cultivated or used by livestock [47].

**Table 2.** AB5 crops during the study period (2019–2020) and its land use.

| Crop Type | Land Use (ha) | % Land Use | Crop Cycle(s) |
| --- | --- | --- | --- |
| Maize | 172 | 41.1 | April-October |
| Alfalfa | 95 | 22.7 | Permanent |
| Barley | 73 | 17.4 | January-June |
| Wheat + Maize | 30 | 7.1 | January-October |
| Olive trees | 22 | 5.3 | Permanent |
| Barley + Maize | 14 | 3.3 | January-October |
| Peas + Maize | 8 | 1.9 | February-October |
| Soybeans | 2 | 0.5 | May-September |
| Grapevine | 2 | 0.5 | Permanent |
| Total | 419 | 100 | |

Table 2 shows that the main cultivated crop was maize, with a total of 224 ha, which was either grown as a single crop (172 ha) or as a double crop after wheat, barley or peas (52 ha). Alfalfa, barley and olives trees were also grown in the area. The main irrigation

system used in the study area was sprinkler irrigation, which is present in 94% of the area [46]. The remaining 6% is drip irrigation, mostly in olive orchards and vineyards.

To establish an approximate estimation of irrigation water use in the sub-basin during the study period, crop water requirements were calculated throughout the growing season using the FAO-56 single crop coefficient [15] methodology. Most growers in the area schedule irrigation using an open-access tool which calculates crop water requirements based on the FAO-56 single crop coefficient assumption (https://ruralcat.gencat.cat/eines/eina-recomanacions-de-reg-agricultura, accessed on 14 March 2022). Therefore, the potential crop evapotranspiration ($ET_c$ in mm) is obtained from Equation (3):

$$ET_c = ET_o \cdot K_c \tag{3}$$

As Allen et al. [4] point out, using different single crop coefficients ($K_c$) to establish a more accurate $ET_c$ result is recommended. Table 3 shows the different lengths of crop development and their $K_c$ coefficient, for each of the crops in the sub-basin.

**Table 3.** Sowing time, lengths of the four distinct growth stages and the growing total in days: initial ($L_{ini}$), development ($L_{dev}$), middle ($L_{mid}$), late ($L_{lat}$) and total (L total). Crop coefficients for the initial stage ($Kc_{ini}$), for the mid-season stage ($Kc_{mid}$) and for the end of late season stage ($Kc_{lat}$) [4]. The sowing dates and the lengths of development were adapted to the Algerri-Balaguer irrigation district characteristics. * Maize FAO 300 is sowed as a secondary crop immediately after barley, wheat or peas.

| Crop | Sowing Time (month) | L ini | L dev | L mid | L lat | L total | $Kc_{ini}$ | $Kc_{mid}$ | $Kc_{lat}$ |
|---|---|---|---|---|---|---|---|---|---|
| Maize (FAO 700) | May | 30 | 45 | 45 | 30 | 150 | 0.3 | 1.2 | 0.6 |
| Maize * (FAO 300) | June | 20 | 30 | 30 | 20 | 100 | 0.3 | 1.2 | 0.6 |
| Alfalfa | March | 30 | 30 | 60 | 60 | 180 | 0.4 | 0.95 | 0.9 |
| Barley | January | 30 | 40 | 40 | 30 | 140 | 0.3 | 1.15 | 0.25 |
| Wheat | January | 30 | 40 | 40 | 30 | 140 | 0.7 | 1.15 | 0.3 |
| Peas | February | 30 | 30 | 30 | 30 | 120 | 0.5 | 1.15 | 0.3 |
| Soybeans | May | 20 | 30 | 30 | 25 | 105 | 0.4 | 1.15 | 0.5 |
| Olive Tree | - | 20 | 60 | 90 | 60 | 230 | 0.65 | 0.7 | 0.7 |
| Grapevine | - | 20 | 50 | 75 | 60 | 205 | 0.3 | 0.7 | 0.45 |

Regarding the theoretical water consumption, water requirements for each crop were estimated from the potential crop evapotranspiration, composed of initial ($ET_c$ ini), middle ($ET_c$ med) and late evapotranspiration ($ET_c$ lat), determined through Equation (3) for each of the three stages ($L_{dev}$ and $L_{mid}$ are in the mid-season stage). Net irrigation requirements (NIR, mm) of each crop were obtained following Equation (4):

$$NIR = ET_c - ER \tag{4}$$

where ER is the effective monthly rainfall (mm), computed using the FAO model, which uses parameters of climate, crop and soil data, is suitable for all kind of situations and has a good accuracy according to Kumar et al. [48]. The model uses Equations (5) and (6) [5]:

$$ER = 0.6 \cdot R - 10 \text{ when } R \leq 75 \text{ mm} \tag{5}$$

$$ER = 0.8 \cdot R - 25 \text{ when } R > 75 \text{ mm} \tag{6}$$

where R is the accumulated monthly rainfall in mm

Finally, gross irrigation requirements (GIR, mm) were calculated as shown in Equation (7), with AE the application efficiency, set as 0.85 in sprinkler irrigation and 0.95 in drip irrigation, in accordance with Martin et al. [49]:

$$GIR = NIR/AE \qquad (7)$$

Different indices were also calculated to assess irrigation management:

- Relative Water Irrigation Index (RWII)

This irrigation performance index was calculated through Equation (8):

$$RWII = (GIR - I) \cdot 100/GIR \qquad (8)$$

where I is the amount of water applied as irrigation (mm). A negative RWII implies that the irrigation applied is higher than the gross irrigation requirements (GIR), calculated using the FAO-56 method.

- Irrigation Efficiency (IE)

Irrigation efficiency was calculated through Equation (9):

$$IE = (I - ID) \cdot 100/I \qquad (9)$$

where irrigation drainage (ID, mm) is the amount of drainage water resulting from applied irrigation.

- Water Efficiency (WE):

Water efficiency was calculated through Equation (10):

$$WE = (I + R - D) \cdot 100/(I + R) \qquad (10)$$

where D is the total amount of drainage water (in mm).

### 2.2.3. Remote Sensing for Evapotranspiration

Actual crop evapotranspiration ($ET_{a\ TSEB}$) was also estimated using the two-source energy balance (TSEB) [50] model with Copernicus-based-inputs [29,51,52]. Further details on the TSEB model scheme can be find at the source code (https://github.com/hectornieto/pyTSEB, last accessed on 10 June 2022) and the original formulation of the model [50]. The Priestley-Taylor (PT) approach [53] was used to run the TSEB with satellite data and to derive directional radiometric temperature (Trad). In order to derive canopy (Tc) and soil temperature (Ts), the TSEB-PT uses a first approximation of canopy latent heat flux, LEc. This includes a first guess of canopy transpiration at the potential rates using an alpha coefficient initially set to 1.26 but subsequently automatically reduced for stress conditions [53,54].

TSEB-PT was run and mosaicked for the entire agricultural area of Lleida (west of Catalonia, Spain) (273,940, 4,573,440; 359,220, 4,653,320 m UTM31 N), and subsequently a subset of the study site was made (Figure 1b). Sentinel-2A and 2B tandem images corresponding to tiles T31 (TBF, TBG, TCF and TCG) were downloaded at Level-2A (L2A; bottom of atmosphere reflectance). The study site was located within the tile T31TCG. The S3 SLSTR sensor was used to obtain thermal data. The land surface temperature (LST) was obtained directly as an L2A product of S3 SLSTR at 1-km resolution. At the study site, a total of 25 cloud-free Sentinel-2 (S2) and 187 cloud-free Sentinel-3 (S3) images were available from 15 October 2019 to 15 October 2020.

Biophysical parameters of the vegetation were estimated from S2 using the biophysical processor available in the SNAP software 8.0 (https://step.esa.int/main/download/snap-download/, last accessed on 12 February 2022) [55]. These parameters were retrieved with the PROSAIL radiative transfer model [56] by simulating reflectance of different variables and training data with a machine learning approach. Among different biophysical

parameters, the leaf area index (LAI) and the fraction of vegetation cover (fc) were used as the inputs of the two-source energy balance (TSEB) model.

The data mining sharpening (DMS) approach [57] was used to disaggregate 1 km S3 LST to 20 m. This methodology relies on the empirical relation that exists between shortwave and thermal data. In DMS, a non-parametric approach is used that considers a full set of shortwave spectral reflectance bands and ancillary data at a coarse resolution. A decision tree machine learning algorithm was subsequently used to relate S3 thermal band imagery to a suite of S2 shortwave spectral reflectance and ancillary data at a coarse resolution, and then to apply it to a fine (shortwave) pixel resolution. Further information about the pyDMS approach and code used in this study is available online (https://github.com/radosuav/pyDMS, last accessed on 17 March 2022).

Meteorological inputs are based on ERA5-Land hourly data on single levels [58] produced by the European Center for Medium Range Weather Forecast and distributed in open access through the Copernicus CDS (https://cds.climate.copernicus.eu/, last accessed on 17 March 2022). The horizontal resolution grid is 50 km. Both instantaneous and daily forcing were derived from ERA5 data. The instantaneous air temperature at 2 m, vapor pressure, dew point temperature at 2 m, wind speed at 10 m and clear-sky solar irradiance at the satellite overpass time were used to drive the ET model. The instantaneous latent heat flux was upscaled to daily water fluxes, expressed in units of mm/day, by multiplying the instantaneous ratio of latent heat flux over solar irradiance by the average daily solar irradiance [33]. Two ancillary sources of data were used: Land Cover maps from the Copernicus Climate Change Service (C3S) (https://cds.climate.copernicus.eu/cdsapp#!/dataset/satellite-land-cover?tab=overview, last accessed on 10 October 2021) at 300 m resolution and the Shuttle Radar Topography Mission (SRTM) DEM. A look-up table of different parameters associated to each crop class was set up according to Guzinski et al. [51].

For dates with cloud coverage and, therefore, without clear satellite imagery, a gap filling approach was applied to obtain $ET_{a\ TSEB}$ data [52]. The approach involves filling dates due to cloud conditions with an assumption that the ratio of reference FAO-56 ET to $ET_{a\ TSEB}$ remains steady for a short-term period.

### 2.2.4. Drainage Water

The drainage water flow was estimated from the water level measurements in the drainage outlet in the sub-basin. The drainage system of the studied sub-basin consists in subsurface polyethylene drain pipes covered with geotextile and gravels with intern diameters from 0.25 to 0.37 m with concrete pipes in the outlet (DN 300–400 mm). The water level was measured using the Hydros21 sensor (Meter Group INC, Pullman, WA, USA) installed specifically for this study. The Hydros21 sensor measures water electrical conductivity corrected at standard temperature of 25 °C (EC at dS/m), water temperature (°C) and water depth (mm). The device was calibrated in the laboratory before installation. The sensor was connected to an Em50G DataLogger (Meter Group Inc., Pullman, WA, USA). Hourly data were downloaded from the digital platform ZentraCloud (Meter Group Inc., Pullman, WA, USA).

Using the water level data, a relationship was stablished to determine the water flow using the Manning equation where we obtained the water velocity in a specific pipe material. Regarding the case study, as the material used for the drainage pipes in the sampling outlet is concrete, the Manning roughness coefficient (*n*) is between 0.012–0.017 [59]. The average value, 0.0145, was selected and the slope (Z) established as 0.0025 m/m. Further information related to flow rate calculations can be found in Appendix A.

In order to assess the drainage performance in the sub-basin, different indices were calculated during the study period with total amount of water (mm).

- Drainage ratio (DR), which corresponds to the leaching fraction, was calculated through Equation (11):

$$DR = (D) \cdot 100/(I + R) \tag{11}$$

- Rainwater drainage ratio (RWDR)

The rainwater drainage ratio was calculated through Equation (12):

$$RWDR = (RWD) \cdot 100/(R) \tag{12}$$

where RWD is the amount of rainwater drained in mm

- Irrigation drainage ratio (IDR)

The irrigation drainage ratio was calculated through Equation (13):

$$IDR = (ID) \cdot 100/(I) \tag{13}$$

where ID is the amount of irrigation water drained in mm.

Therefore, D is the sum of RWD and ID.

On the other side, the DR was compared with the leaching requirement (LR), estimated as Equation (14) [11], which will allow us to assess the surplus in water drainage in the study area, where $EC_w$ is the average electrical conductivity of irrigation water (dS/m) and $EC_e$ the electrical conductivity of saturated paste extract of the soils in the study area.

$$LR = EC_w/(5 \cdot EC_e - EC_w) \tag{14}$$

- HEC-HMS model

The Hydrologic Engineering Center-Hydrological Model System (HEC-HMS) from the US Army Corps of Engineers [60] was used to assess the fraction of drainage water that comes from rainfall. This is a generalized modelling system capable of representing different watersheds. These models are constructed by separating the hydrological cycle into manageable pieces [61], with each piece having to be defined by the user [62]. Several studies have been carried out to estimate hydrological behavior using this methodology, in small forests [63,64] and large watersheds [65–67].

- Sub-basin model

Table 4 lists the six parameter methods used in the HEC-HMS model. A simple canopy method was used with evapotranspiration during dry periods, a $K_c$ value of 1 and the reference evapotranspiration values ($ET_o$) obtained in the Albesa agrometeorological station. The Soil Conservation Service (SCS) curve number (CN) was estimated for the SCS-CN loss method following the methodology established in Spain for "surface drainage of highways" reflected in Spanish legislation Order FOM/298/2016 [68], considering the soils of the study zone as silty clay loam and belonging to soil group C. The relation between initial abstraction ($I_a$) and the CN is given by Equation (15). $I_a$ is defined as the losses before runoff begins, related with the potential maximum retention after runoff begins (S), as shown in Equation (16) [69].

The Clark unit-hydrograph procedure was selected as the transform method, obtaining the time of concentration ($t_c$ in hours) with the methodology of [68], following Equation (17) and the storage coefficient (SC) following Equation (18) [62].

The constant monthly baseflow method was employed, establishing a baseflow of 0.005 m$^3$/s during the whole year, considering that value to be the deep percolation of rainfall events.

**Table 4.** HEC-HMS methods used in the study and their values.

| Sub-Basin Model | | | |
|---|---|---|---|
| **Parameter Method** | **Method Used** | **Values** | |
| Discretization model | No discretization | - | - |
| Canopy method | Simple canopy | Initial storage (%) | 33 |
| | | Max storage (mm) | 40 |
| | | Evapotranspiration | Only in dry periods |
| | | Uptake method | Simple |
| Surface method | None | - | - |
| Loss method | SCS curve number | $I_a$ (mm) | 14 |
| | | CN | 79 |
| Transform method | Clark unit hydrograph | Method | Standard |
| | | Time of concentration (h) | 3.12 |
| | | Storage coefficient (h) | 5.8 |
| | | Time-Area method | Default |
| Baseflow method | Monthly constant | Monthly value ($m^3/s$) | 0.005 |

- Meteorological model

A meteorological model comprising precipitation and evapotranspiration values was used in the methodology. Data were obtained on an hourly basis from the Albesa agrometeorological station (XAC, Servei Meterorològic de Catalunya) during the period from 15 October 2019 at 00:00 h to 14 October 2020 at 23:00 h.

$$CN = 25{,}400/(254 + (I_a \text{ (mm)}/0.2)) \tag{15}$$

$$I_a = 0.2 \cdot S \tag{16}$$

$$t_c = 0.3 \cdot L_c^{0.76} \cdot J_c^{-0.19} \tag{17}$$

where $L_c$ is the maximum course length (6.9 km) and $J_c$ the slope of the course (0.01 m/m).

$$SC/(t_c + SC) = 0.65 \tag{18}$$

2.2.5. Statistical Analysis

The irrigation data were analyzed using a univariate analysis of variance (ANOVA). Tukey's HSD was used to compare the amounts of water used ($p \leq 0.05$). The statistical analyses were carried out using the JMP Pro 16.0.0 software package (SAS Institute Inc., Cary, NC, USA).

**3. Results**

*3.1. Amount of Irrigation Water Applied*

A total of 40 fields with their corresponding irrigation hydrants were monitored at the beginning and end of the study period. These data represent the total amount of irrigation water applied by farmers and reflects their irrigation practices. The results are shown aggregated according to land use (Table 5). The total amount of water used for irrigation was 2,573,779 $m^3$, which represents an average of 6142 $m^3$/ha. The largest volume was used to irrigate maize (1,287,119 $m^3$), with an average of 7509 $m^3$/ha, followed by alfalfa (660,288 $m^3$) with an average of 6598 $m^3$/ha. The combination of barley + maize had the highest use of irrigation water per hectare with an average of 7879 $m^3$/ha.

**Table 5.** Average irrigation and total water applied of the different crops in the AB5 basin during the study period.

| Crop | Average Irrigation (m³/ha) | Standard Deviation (m³/ha) | Total Water Applied (m³) | Number of Fields (*n*) |
|---|---|---|---|---|
| Maize | 7574 [A] | 2294 | 1,287,119 | 13 |
| Alfalfa | 6598 [A] | 1600 | 660,288 | 3 |
| Barley | 2371 [B] | 1143 | 219,100 | 5 |
| Wheat + Maize | 6438 [A] | 143 | 191,815 | 3 |
| Olive | 2320 [B] | 3327 | 54,266 | 8 |
| Barley + Maize | 7879 [A] | 851 | 105,652 | 4 |
| Peas + Maize | 5105 [A,B] | 292 | 40,176 | 2 |
| Soybeans | 6067 [A,B] | 0 | 13,832 | 1 |
| Grapevine | 680 [B] | 0 | 1531 | 1 |

Means not connected by the same letter are significantly different according to Tukey's HSD test ($p \leq 0.05$).

### 3.2. Estimated Crop Water Irrigation Requirements

The estimated potential crop evapotranspiration ($ET_c$), net and gross irrigation requirements per crop, applying the FAO-56 method [4], are shown in Table 6.

**Table 6.** Potential crop evapotranspiration ($ET_c$), net irrigation requirements (NIR) and gross irrigation requirements (GIR) for the crops in the AB5 sub-basin for a given period.

| Crop | $ET_{c\ ini}$ (mm) | $ET_{c\ mid}$ (mm) | $ET_{c\ lat}$ (mm) | $ET_c$ (mm) | NIR (mm) | GIR (mm) |
|---|---|---|---|---|---|---|
| Alfalfa | 26 | 352 | 293 | 672 | 556 | 654 |
| Maize (FAO 700) | 41 | 571 | 31 | 643 | 591 | 696 |
| Maize (FAO 300) | 45 | 391 | 62 | 499 | 484 | 569 |
| Barley | 5 | 224 | 34 | 264 | 109 | 129 |
| Wheat | 13 | 245 | 41 | 298 | 144 | 169 |
| Peas | 22 | 174 | 41 | 236 | 134 | 158 |
| Olive Tree | 43 | 488 | 36 | 567 | 332 | 391 |
| Soybeans | 54 | 324 | 76 | 454 | 403 | 474 |
| Grapevine | 20 | 260 | 147 | 426 | 191 | 225 |

According to the methodology carried out, maize and alfalfa were the crops with the highest GIR (696 mm and 654 mm during their cycle, respectively), while the lowest GIR were for barley, peas and wheat.

Finally, a summary of the total GIR for each crop is shown in Table 7. The total amount of estimated irrigation requirements at the AB5 basin was 2,389,826 m³.

Therefore, the difference between measured and estimated irrigation requirements using the FAO-56 soil-water balance approach (GIR) was 183,953 m³. This result indicates that, on average, the actual amount of irrigation water applied was 7.7% higher than the theoretical estimation using the FAO-56 approach (RWII). Alfalfa has the tightest RWII and it is difficult to improve its water management (−0.8%). On the other hand, maize (−8.9%) and barley + maize (−12.9%) had lower RWII values but also a potential improvement in irrigation management. In addition, barley had the highest overirrigation with an RWII of

−84.5%, and grapevine (69.8%) and olive (40.7%) the highest underirrigation during the study period.

**Table 7.** Land use, gross irrigation requirements per hectare, average irrigation, total gross irrigation requirements and relative water irrigation index (RWII) for the crops in the AB5 basin during the study period.

| Crop | Land Use (ha) | GIR (m³/ha) | I (m³/ha) | GIR (m³) | RWII (%) |
|---|---|---|---|---|---|
| Maize | 172 | 6958 | 7574 | 1,196,756 | −8.9 |
| Alfalfa | 95 | 6544 | 6598 | 621,669 | −0.8 |
| Barley | 73 | 1285 | 2371 | 93,826 | −84.5 |
| Wheat + Maize | 30 | 7388 | 6438 | 221,644 | 12.9 |
| Olive | 22 | 3910 | 2320 | 86,023 | 40.7 |
| Barley + Maize | 14 | 6980 | 7879 | 101,128 | −12.9 |
| Peas + Maize | 8 | 7277 | 5105 | 58,215 | 29.8 |
| Soybeans | 2 | 4737 | 6067 | 9473 | −28.1 |
| Grapevine | 2 | 2251 | 680 | 4503 | 69.8 |

### 3.3. Drainage Water

During the study period, data were measured hourly in the outlet of the AB5 sub-basin. Figure 3 shows the hourly evolution of water flow and rainfall. The dynamics of drainage water flow ($Q_d$) shown in Figure 3 are associated with a continuous increase over time due to the amount of irrigation applied, with fluctuations associated with rainfall events, including the rains of 22 April and 7 May 2020 (reaching a peak of water discharge higher than 250 m³/h). After 28 August $Q_d$ decreases as less irrigation water is applied. A monthly data summary is shown in Table 8. The total water volume discharged in the outlet during the study period was 862,142 m³, with the months with the most important discharge volume being May (associated mainly to rainfall events), and July and August (associated to the months where more irrigation water is applied).

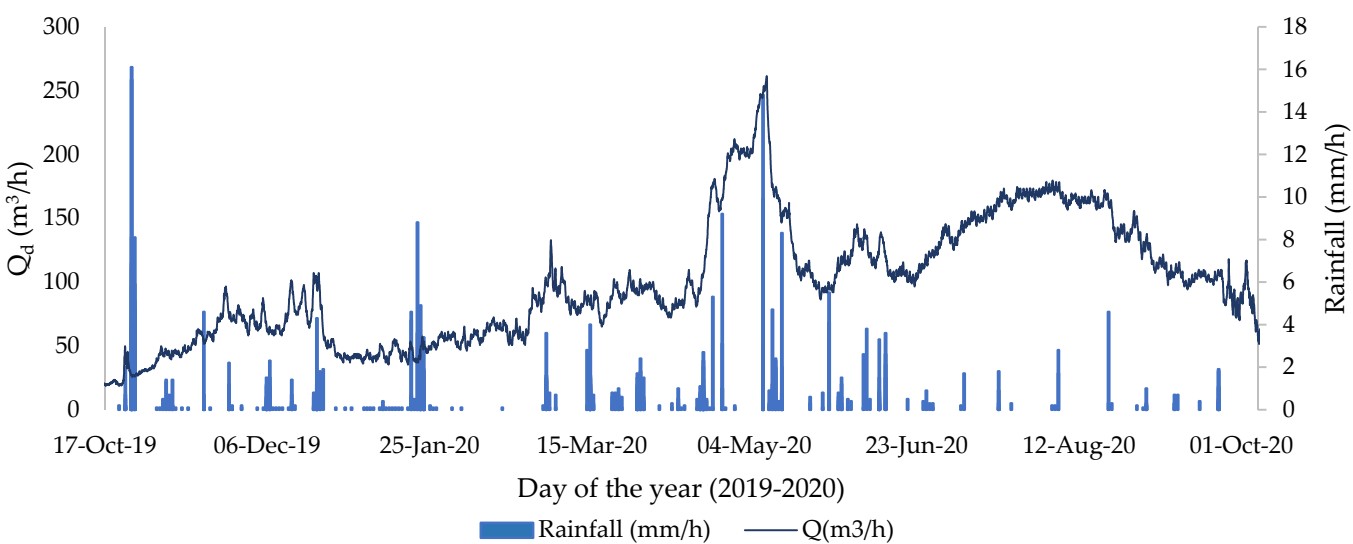

**Figure 3.** Drainage water flow ($Q_d$, m³/h) in the outlet and rainfall (mm/h) episodes during the study period (15 October 2019 to 15 October 2020).

**Table 8.** Average water level of the monitored outlet (y, mm), average velocity (V, m/s) during each month, average drainage water flow ($Q_d$, $m^3$/s) daily drainage average water flow ($Q_d$, $m^3$/day) and drainage discharge ($Q_d$, mm/day), total monthly flow ($Q_m$, $m^3$/month) and total monthly rainfall (R, mm).

| Year | Month | y (mm) | V (m/s) | $Q_d$ ($m^3$/s) | $Q_d$ ($m^3$/day) | $Q_d$ (mm/day) | $Q_m$ ($m^3$/month) | R (mm) |
|---|---|---|---|---|---|---|---|---|
| 2019 | October | 77 | 0.44 | 0.007 | 628 | 0.148 | 9422 | 98 |
| | November | 115 | 0.56 | 0.016 | 1391 | 0.327 | 41,716 | 21 |
| | December | 124 | 0.58 | 0.019 | 1617 | 0.380 | 50,112 | 42 |
| 2020 | January | 103 | 0.52 | 0.013 | 1109 | 0.261 | 34,367 | 72 |
| | February | 122 | 0.57 | 0.018 | 1541 | 0.363 | 44,695 | 1 |
| | March | 146 | 0.63 | 0.025 | 2153 | 0.507 | 66,745 | 54 |
| | April | 176 | 0.68 | 0.035 | 3030 | 0.713 | 90,886 | 83 |
| | May | 194 | 0.71 | 0.041 | 3538 | 0.832 | 109,674 | 78 |
| | June | 170 | 0.68 | 0.033 | 2832 | 0.666 | 84,951 | 40 |
| | July | 200 | 0.73 | 0.043 | 3732 | 0.878 | 115,695 | 7 |
| | August | 201 | 0.73 | 0.044 | 3777 | 0.889 | 117,079 | 13 |
| | September | 158 | 0.66 | 0.029 | 2502 | 0.589 | 75,052 | 8 |
| | October | 118 | 0.56 | 0.017 | 1466 | 0.345 | 21,849 | 18 |

HEC-HMS Model

The total cumulative rainfall in the entire sub-basin (4.25 $km^2$) during the study period (15 October 2019 to 15 October 2020) was 2,264,825 $m^3$. The water infiltrated into the soil was estimated through the HEC-HMS model as 2,056,575 $m^3$ (90.8%). The deep percolation (base flow) was estimated as 157,675 $m^3$ (7.7% of infiltrated water) and the accumulated water in the soil was 1,898,900 $m^3$ (92.3% of the infiltrated water). Water that was not infiltrated into soil was considered as runoff water (208,250 $m^3$). As the runoff water goes directly to the drainage network, it can be stated that the contribution of rainfall to the drainage flow is, in total, 365,925 $m^3$ (deep percolation + direct runoff).

According to the model used, the contribution of rainfall to the total amount of drainage water in the sub-basin AB5 was estimated to be 42.4%.

*3.4. Actual Evapotranspiration (ET$_a$)*

The monthly mean values of remotely sensed estimated actual evapotranspiration (ET$_{a\ TSEB}$) during the study period are shown in Table 9. The total estimated amount of cumulative ET$_{a\ TSEB}$ for the entire sub-basin was 4.86 $hm^3$. Figure 4 shows the cumulative ET$_{a\ TSEB}$ values for the study period for each of the fields that make up the sub-basin.

The total average ET$_{a\ TSEB}$ value during the study period was 1144 mm. The months with the highest values were August with 161 mm, followed by May and July. The winter months of December, January and February had the lowest levels of ET$_{a\ TSEB}$. For October, we only considered the period from the 1st to the 15th, with the total value for these days being 38 mm for 2019 and 37 mm for 2020.

A comparison is also made in Table 9 of the three ETs studied. ET$_o$ and ET$_{a\ TSEB}$ values are very similar, with a yearly difference of just 3%. Total ET$_{c\ FAO\ 56}$ is lower, mainly because it is estimated as the ET during the crop cycle and is 38% less than the ET$_{a\ TSEB}$.

**Table 9.** Monthly mean values of $ET_o$, $ET_{c\ FAO56}$ and $ET_{a\ TSEB}$ in mm during the study period.

| Year | Month | $ET_o$ | $ET_{c\ FAO\ 56}$ | $ET_{a\ TSEB}$ |
|------|-------|--------|-------------------|----------------|
| 2019 | October | 64 | | 39 |
| | November | 34 | | 43 |
| | December | 18 | | 41 |
| 2020 | January | 18 | 4 | 46 |
| | February | 44 | 13 | 58 |
| | March | 66 | 30 | 94 |
| | April | 85 | 50 | 114 |
| | May | 136 | 65 | 154 |
| | June | 150 | 115 | 106 |
| | July | 174 | 183 | 133 |
| | August | 152 | 159 | 161 |
| | September | 104 | 83 | 117 |
| | October | 65 | | 37 |
| Total | | 1110 | 702 | 1144 |

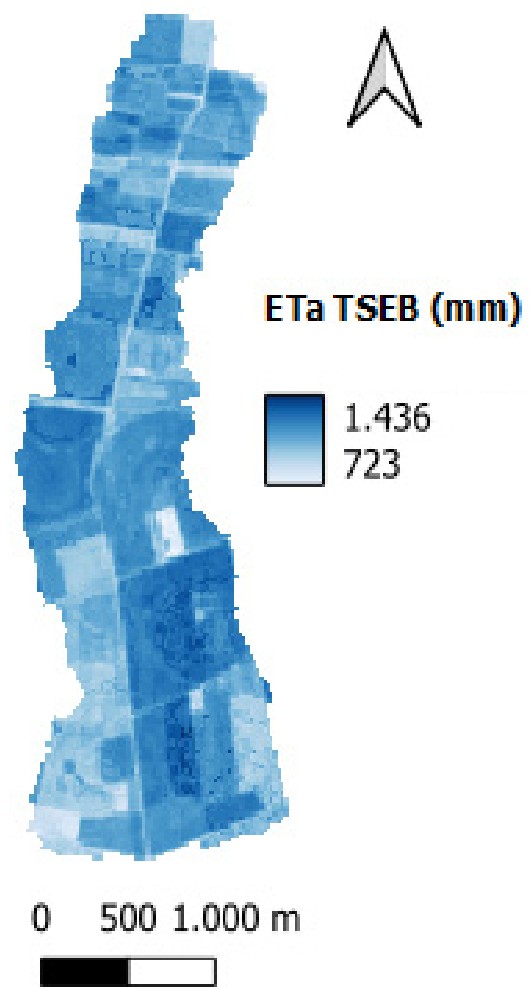

**Figure 4.** Cumulative $ET_{a\ TSEB}$ during the study period in the AB5 sub-basin.

*3.5. Water Balance at the Sub-Basin Level*

The water balance data measured at the sub-basin level were irrigation (2,573,779 m$^3$), rainfall (2,264,825 m$^3$) and drainage (862,142 m$^3$). Using these values in the water balance equation, the results show an ET$_{a\,B}$ value of 3,976,462 m$^3$. In comparison, the remotely sensed ET$_{a\,TSEB}$ was estimated as 4,863,604 m$^3$, a 22.3% higher.

On the other hand, the total amount of drained water (D) corresponds to 862,142 m$^3$, being DR 18%. Moreover, ID corresponds to 496,217 m$^3$, being IDR 19.3% and RWD to 365,925 m$^3$ being RWDR 16.2%.

Therefore, we could state that the 57.6% of the drainage water is ID and the 42.4% remaining is RWD.

The leaching requirements (LR) in the study area for an EC$_w$ of 0.37 dS/m and EC$_e$ of 2dS/m is 3.8%, a significantly lower value than the obtained IDR.

**4. Discussion**

This study allowed an understanding of the dynamics of drainage in irrigated lands and an estimation of the different proportions of drainage water corresponding to rainfall and irrigation. In addition, we quantified the different variables of the soil water balance for the entire sub-basin. The set of results provides us with greater knowledge about the possibilities of improving irrigation in an area where we can affirm that, in general, an excellent management of irrigation water is being carried out.

The irrigation district has an administrative concession from the Ebro Basin Authority (www.chebro.es, last accessed 25 June 2022) of 48 hm$^3$ per year. Therefore, the unitary concession is equivalent to 6000 m$^3$/ha. However, as only 6858 ha are under irrigation, water use is slightly higher than the unitary concession. While average annual rainfall in the study area is 378 mm, during the study period it was 29% higher (533 mm). A rainy spring in 2020 conditioned the amount of irrigation applied in the sub-basin to a total of 2,573,799 m$^3$, which represents 6142 m$^3$/ha on average. As water requirements of some crops are below this average, irrigation water use efficiency can be mostly improved in those crops, or in fields where water use was above the average.

The high energy cost of irrigation water may influence the adjusted use of irrigation water. This may have led to some of the results obtained, with irrigation of the most water-intensive crops having relatively low standard errors.

The approach proposed by FAO-56 is the most widely used method for irrigation water management [70]. The crop coefficients used were for non-stressed, well-managed crops in a sub-humid climate. It is striking that the actual irrigation practice was found to be close to the theoretical needs, with an RWII of −7.7%. Crops that occupy the largest area (maize and alfalfa) are those with the RWII closest to zero. Some crops, such as barley with its high and negative RWII, appear to have been over-irrigated for the meteorological conditions of the year.

Some authors have used the irrigation performance index (IPI), which is the ratio between NIR and I [6,45]. In the study area, the IPI was 79.3%, which was slightly lower than what is considered high-quality irrigation management (100 ± 15) [71]. Traditionally, irrigation efficiency, IE, is a measure of the amount of water that is beneficially used divided by the amount of water applied. In the present study, the IE value was calculated as 80.2%, slightly lower than the values that would be considered optimal, this may be due to the fact of the overirrigation in the barley farms, which correspond to the 17% of the cropped land in the basin; however, the WE index was estimated as 82.2%.

The drainage ratio (DR) was 18%. This represents a significant amount of water that leaves the perimeter of the irrigation district, which is potentially reusable, the 57% of this water corresponds to irrigation water and the remaining 43% to rainwater. The purpose of such indices is generally to allow comparison between irrigation districts or sub-basins, but it also enables the establishment of a benchmark to know the possibilities of improving irrigation water management. During the studies carried out in others sub-basins of the irrigated area in 2006 and 2008 [34], with less intensive monitoring and monthly data

sampling, a DR between a 7% and 18% was estimated. In that study, results showed a drainage discharge during August between 0.11 and 0.396 mm/day, which are lower than the results obtained in the present study. This may be due to the low sampling and less portion of irrigated land, and mainly orchard trees, in the sub-basins.

If we compare the IDR with the LR, we can state that the leaching requirements were highly accomplished, with a surplus of almost five time more water used for leaching than the theoretical required ones. This means that the improvement of irrigation management ant the reduction in water loss may go through the direction on reducing the actual irrigation drainage ratio (IDR).

One of the key aspects of this work has been to establish the proportion of drainage water that results from rainfall and the proportion that results from irrigation water. The HEC-HMS model provides an answer that is widely used in both civil engineering and natural systems. The result obtained in the present work is considered suitable for the purpose of this study and is within the order of magnitude of other works [16,72].

The rainwater drainage ratio (RWDR) and irrigation drainage ratio (IDR) were 16.2% and 19.3%, respectively. Although it can be considered that there is practically no margin for action to minimize the deep percolation of rainfall, aspects such as irrigation frequency and the amounts to be applied would allow improvement of the IDR. On the other hand, the variation of water content in the soil ($\Delta$SW) has been considered as 0 during the study period due to the low impact of it on the total balance in an irrigated sub-basin [14] during an entire hydrological year.

A difference of 22.3% was observed between the $ET_{a\,B}$ and the $ET_{a\,TSEB}$, with the latter having the higher value. This discrepancy may possibly be attributable to the fact that the water balance is closed annually and therefore many things throughout the season were not monitored. Soil evaporation due to heavy rainfall events, transpiration due to the implementation of cover crops in the inter-row during spring, or bare soil during winter months are just a few examples that may explain the bias between the two methodologies. This is in agreement with other studies, which argue that interannual variability in precipitation may explain part of the error [73,74]. In addition, in the case of cereals, $ET_{c\,FAO\,56}$ is estimated during the development period. However, after reaching physiological maturity, when irrigation is no longer required, evapotranspiration is still occurring but without interest for water management. This is seen during the months of May (for barley, wheat and pea), and September and October (for maize). In addition, the single-crop coefficient methodology used [4] could lead to underestimation of the evaporation from soil. An alternative method to avoid this discrepancy would be the FAO-56 dual crop coefficient method [4,75]. Despite this, it is important to point out that the TSEB modelling approach using Copernicus-based inputs has been successfully used and validated in multiple land covers and climatological conditions, achieving an average RMSE of 80–90 W $\cdot$ m$^{-2}$ for sensible and latent heat fluxes, respectively [51]. Therefore, in future studies it may be interesting to explore the assimilation of $ET_{a\,TSEB}$ in water balance models at basin level, in order to estimate other components of the water balance that are difficult to estimate such as drainage or irrigation water applied.

The closure of the water balance in the sub-basin based on the proposed methodology has provided reasonable results of the same order of magnitude as other studies in semi-arid areas and under similar crop conditions [16,72,76].

## 5. Conclusions

The research findings of this study provide evidence that, although water efficiency in the study area is high, irrigation management can still be improved. Key to this potential improvement is the careful monitoring of irrigation and drainage, as well as cooperation between farmers, technicians and researchers. In addition, the use of in situ irrigation and drainage data collected in an automated manner along with remote sensing can be a tool in the future to achieve the goal of improved irrigation efficiency. In this paper, we proposed a model to estimate rainwater and irrigation water drainage. However, works that are

already underway based on the isotopic study of hydrogen and oxygen in irrigation and rainfall water will allow greater accuracy in this respect and facilitate a more detailed assessment of the model. With the present methodology, we estimated a loss of water through drainage (DR) of 18%. The present study permitted to estimate what percentage of the total drainage water comes from rainwater (42.4%) and what percentage comes from irrigation (57.6%). The only way to reduce drainage losses is through improvement of irrigation management. The available margin of improvement is between 19.3% of the present IDR and the 3.8% estimated with the theoretical LR model.

**Author Contributions:** Conceptualization, V.A. and J.M.V.; methodology, V.A.; software, V.A. and J.B.; validation, M.P., J.B. and J.M.V.; formal analysis, V.A.; investigation, V.A.; resources, V.A. and J.B.; data curation, V.A. and J.B.; writing—original draft preparation, V.A.; writing—review and editing, J.B., M.P. and J.M.V.; visualization, V.A.; supervision, M.P. and J.M.V.; project administration, J.M.V.; funding acquisition, J.M.V. All authors have read and agreed to the published version of the manuscript.

**Funding:** This research was performed under the PCI2020-112030 funded by Agencia Estatal de Investigación, Ministerio de Ciencia e Innovación: MCIN/AEI/10.13039/501100011033 and by the European Union NextGenerationEU/PRTR and supported by the IDEWA project (ANR-19-P026-003).

**Data Availability Statement:** Not applicable.

**Acknowledgments:** We are grateful to the technicians and farmers of the Algerri-Balaguer irrigation district for their huge collaboration. We are also thankful to Daniela Álvarez for their assessment in geological description of the area.

**Conflicts of Interest:** The authors declare no conflict of interest.

### Appendix A

Manning equation to calculate velocity in a specific material was applied through Equation (A1). Moreover, velocity values of water drainage were measured in situ several times during the growing season. In these measurements we obtained velocities of 0.6 m/s with a water level of 130 mm in the studied outlet and 1.03 m/s with a water level of 175 mm in another outlet, which resulted in 0.0145 and 0.019 Manning number ($n$), respectively. Although the pipes in the AB5 sub-basin are mainly polyethylene, in the final point of the outlet, where water is monitored, the material is concrete, being realistic the Manning number obtained.

$$V \text{ (m/s)} = 1/n \cdot r_h^{2/3} \cdot Z^{1/2} \tag{A1}$$

The hydraulic radius ($r_h$) can be defined with Equation (A2), the wet area (A) with Equation (A3) and the wet perimeter (WP) with Equation (A4), with r being the radius of the tube (m), $\phi$ the diameter of the tube (m), $\vartheta$ the angle shown in Figure A1 and defined by Equation (A5) and "y" the water level (m) obtained from the Hydros21 sensor:

$$r_h \text{ (m)} = \text{wet area (m}^2\text{)}/\text{wet perimeter (m)} \tag{A2}$$

$$A \text{ (m}^2\text{)} = (r^2 \cdot (\vartheta - \sin \vartheta))/2 \tag{A3}$$

$$WP \text{ (m)} = \vartheta \cdot R \tag{A4}$$

$$\vartheta = 2 \cdot \text{arcos} ((1 - 2 \cdot y)/\phi) \tag{A5}$$

The hourly water flow (Q, m$^3$/s) was determined using Equation (A6), where V is the velocity (m/s) and S the pipe section (S = $\pi \cdot r^2$):

$$Q \text{ (m}^3\text{/s)} = V \cdot S \tag{A6}$$

In the case study, the radius of the pipe was 0.185 m (diameter 0.37 m).

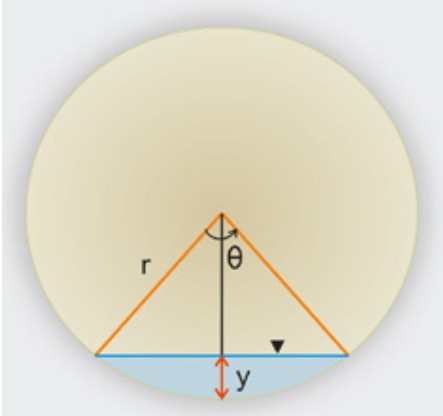

**Figure A1.** Geometric definition of radius (r), angle ($\vartheta$) and water level (y) of a circular perimeter pipe.

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
