# Peer review of "Understanding Drainage Dynamics and Irrigation Management in a Semi-Arid Mediterranean Basin"

_water, doi:10.3390/w15010016_

Round 1

Reviewer 1 Report

The manuscript is very suitable this “Water MDPI – Journal”. After addition all my notes, I recommend this manuscript ID water-2070453 for publication

Page 2.

Chapter 2.1. Study area description

I miss, in the corresponding place at the text, following:

Natural conditions

-        Fundamental description of hydrogeology (geology) information, especially stratification and description of layers, position of the low permeable (impervious) layer (if any)

-        Fundamental description of soils (surface layers), the results of soil hydrology investigations, the value of hydraulic conductivity (conductivities), drainable pore space, soils character from the permeability point of view, simply: how permeable /low, middle permeable/ are the soils

-        Hydrology of rivers, e.g. how is the long term annual flow rate in Noguera Ribacorgana River (with support of basins and reservoirs) in connection with irrigation? Is crop water requirements satisfactory? How were planned (calculated) crop water requirements? Is water supply for pigs, poultries, cattle farming, etc., etc. satisfactory? How was projected (quantify) water supply for cattle farming?     

-        Precipitation and evapotranspiration hydrology – how long is time series for determination of average annual rainfall precipitation, respectively evapotranspiration? How long is time series for determination of average air temperature?

Drainage

-        I miss the drain spacing of ditches (distance between individual ditches /open drains/), how are their values and where the design of drainage system comes from?

-        I miss the drain spacing of subsurface pipes (distance between individual pipes /subsurface pipe drainage system/), how are their values and where the design of subsurface drainage system comes from?

-        I miss water table level in drained area, how was the water table level before (and after) drainage system -

-        I miss drain discharge (annual /daily/) usually in (litres/sec/ha) or (mm/days); you have series of measures 1998 – 2017 resp. 2006 – 2008 (line 99 – 104), did you measure drain discharge with corresponding water table level in this period?

-        Did you measure precipitation in this period (1998 – 2017 resp. 2006 – 2008)

-        Did you measure irrigation in this period (1998 – 2017 resp. 2006 – 2008)

-        You mentioned Chezy’s Equation with Manning formula (page 18, Appendix A) to calculate velocity of the drain pipes. Can you specify the results? Means concrete velocity in the drain pipe. Do not be come about drain pipes clogging?  

Page 6 – 7, Table 3 (Sowing dates); I miss units

Most equations (formulas) do not content the units. E.g. formula (10)

WE = (I + R – D) x 100/ (I + R)

Units of WE, Water Efficiency??, in % ok; but units of I – total irrigation water (m3? litres? mm?

Units of R – total rainfall (mm??)

Units of D – total amount of drainage water (mm? litres/sec/ha or mm/day??

Etc., etc …..

Equations (11), (12); what is the reason of “P”? Where can I get it?

Page 13, Figure 2

Q (m3/h) is water flow, total water flow or drainage water flow (page 12, line 439 – 444)?

Where is drawn the value of irrigation water? 

The range between individual days (day of the year – X axis) is very unobvious, e.g. 04-05 means the fourth of May?  

Figure 2 demonstrates “Day of the year”, but what year? One certain (concrete) year? Which one? Typical (representative) year from certain period? Please, can you specify this?

Page 13, Chapter 3.3.1. HEC-HMS model (line 456 – 463) From what period are there the data mentioned in HEC-HMS model? Long term annual?   

Author Response

First of all, thank you for the useful comments, we have done our best to clarify all the issues raised regarding the manuscript. Find attached the new version of the manuscript in which we added some information in order to clarify all the observation from you and your colleagues and improve the quality of the manuscript, and point-by-point responses to your comments.

Natural conditions

-        Fundamental description of hydrogeology (geology) information, especially stratification and description of layers, position of the low permeable (impervious) layer (if any)

      Thank you for this comment, it was indeed lacking a more detailed geological information. We added L117-151, a precise description from geology and hydrogeology in the area, from general to specific to the studied sub-basin, in order to contextualize the reader the geological context of the area.

-        Fundamental description of soils (surface layers), the results of soil hydrology investigations, the value of hydraulic conductivity (conductivities), drainable pore space, soils character from the permeability point of view, simply: how permeable /low, middle permeable/ are the soils

      Thank you for this observation, we added a detailed soil description of the main soils in the study area, with the hydraulic properties obtained from the studies cited in the paper in L151-165 and Table 1 L170 in the revised version

-        Hydrology of rivers, e.g. how is the long term annual flow rate in Noguera Ribacorgana River (with support of basins and reservoirs) in connection with irrigation?

Thank you for this observation. This is an interesting point what we missed to address. Noguera Ribagorçana river is a highly regulated river with four dams upstream the study area (Baserca, Escales, Canelles and Santa Ana dams, from upstream to downstream). Water used for irrigation is mainly supplied from the Santa Ana dam, not only to the Algerri-Balaguer irrigation district (the studied area), but also to two other irrigation districts, Canal de Aragón-Cataluña district (105,000 ha) and Piñana district (12,000 ha). When water is demanded by any of the districts downstream Santa Ana, water (managed by Condeferación Hidrográfica del Ebro- Ebro Basin Authority) is released in addition to the constant ecological flow of the river (averaged to 6.8 m3/s), with the aim of not disturb and modify the environment downstream. In the case of the Algerri-Balaguer district, the maximum water demand is 4.8 m3/s. In order to address this issue, we added details in L189-194: Water used in the irrigation district is pumped 32 m from the Noguera Ribagorçana River a highly regulated river with four dams, belonging to the Ebro River basin, north of the city of Lleida (NE Spain), and south of the Barbastro-Balaguer anticline (Fig. 1). Water demanded by the irrigation district, with a maximum flow of 4.8 m3/s, is released from the Santa Ana dam at any time of the year, a water reservoir of 236 hm3 located 380 meters above sea level (m.a.s.l.) (UTM X: 797086.1, UTM Y: 4642949.4), in addition to the ecological flow of 6.8 m3/s (on average) from the river.  

Is crop water requirements satisfactory? How were planned (calculated) crop water requirements? Is water supply for pigs, poultries, cattle farming, etc., etc. satisfactory? How was projected (quantify) water supply for cattle farming?    

Thank you for this comment, as we point in our discussion L615-616, the irrigation district has an administrative concession of 48 hm3/year for a maximum of 8,000 ha (corresponding to a average consumption of 6000 m3/ha). As only 6,858 ha, the water requirements are largely covered for crops either for cattle farms. The irrigation district made no planification on crop or cattle water requirements during implementation as they had guaranteed the concession of 48 hm3/year. In order to address this topic, we modified L617-618: However, as only 6,858 ha are under irrigation, water use is slightly higher than the unitary concession.

-        Precipitation and evapotranspiration hydrology – how long is time series for determination of average annual rainfall precipitation, respectively evapotranspiration? How long is time series for determination of average air temperature?

      Thank you for the observation. We changed the order of the cite (43) (Dades agrometeorològiques - Ruralcat (gencat.cat)) from L233 to L226 to clarify that all air temperature, rainfall and ET data are obtained from this source during the indicated years (2000-2021), as well as during the study period.

Drainage

-        I miss the drain spacing of ditches (distance between individual ditches /open drains/), how are their values and where the design of drainage system comes from?

      Thank you for this comment. In order to clarify this and other comments we changed L105-L106: In order to implement irrigation, actions on land consolidation, the installation of a pressurized irrigation network and a drainage network, and the establishment of Special Protection Areas (SPAs) were developed from 1998 to 2017 by the public regional infrastructure company (Infraestructures.gencat.cat).

      Information related to main ditches can be found in the work project reports, in the case of the studied sub-basin, all drains are subsurface drains

-        I miss the drain spacing of subsurface pipes (distance between individual pipes /subsurface pipe drainage system/), how are their values and where the design of subsurface drainage system comes from?

      Thank you for the observation on this issue. Related to subsurface pipes installed in farms, we have no data on their spacing or dimensions, as they were installed over time with the evolution of irrigation by the owners of the farms. Also, the design of them is not known for us.

      In the case of the subsurface pipes installed by the public regional infrastructure company during the process of irrigation implementation, we do have the data of their dimensions but not the documents with their longitudes and distance between pipes. In 2.2.4. L378-380 we added: The drainage system of the studied sub-basin consists in subsurface polyethilene drain pipes covered with geotextile and gravels with diameters from 0.25 to 0.37 m with concrete pipes in the outlet.

-        I miss water table level in drained area, how was the water table level before (and after) drainage system –

      Related to this issue, we modified L206-207: The main drainage network is designed to evacuate excess irrigation and rainfall water and avoid a rise in the water table, found at depths from 0.5 to 2.15 m in some occasional sports near the natural drainage system during the soil survey study conducted in the area during 1991 [37].

-        I miss drain discharge (annual /daily/) usually in (litres/sec/ha) or (mm/days);

      Indeed, we miss it also, as we have the data at hourly basis, we added it in the paper in the results of drainage in mm/days in table 8 L501

you have series of measures 1998 – 2017 resp. 2006 – 2008 (line 99 – 104), did you measure drain discharge with corresponding water table level in this period?

Regarding the measures from 1998-2017, we are sorry for the misunderstood, in L99-104 we meant measures (actions) on building and construction. To avoid further misunderstandings, we changed the sentence to L105-108: In order to implement irrigation, actions on land consolidation, the installation of a pressurized irrigation network and a drainage network, and the establishment of Special Protection Areas (SPAs) were developed from 1998 to 2017 by the public regional infrastructure company (Infraestructures.gencat.cat) .

Regarding 2006-2008 period, we did not enter in detail on that study because it was conducted in a different sub-basin of the irrigation district, notwithstanding, we have data on discharge (L/s) during certain days, one per month, as we did not have installed any water level sensor and we are adding in the discussion,

-        Did you measure precipitation in this period (1998 – 2017 resp. 2006 – 2008)

      The automatic meteorological station in Albesa (ref 43) is recording data since 2000, and we did obtain the average precipitation during the period 2000-2021 and in the study developed during 2006-2008 we did take data from that period.

-        Did you measure irrigation in this period (1998 – 2017 resp. 2006 – 2008)

      Also during the 2006-2008 period the irrigation was measured, but not at farm scale as this study did.

-        You mentioned Chezy’s Equation with Manning formula (page 18, Appendix A) to calculate velocity of the drain pipes. Can you specify the results? Means concrete velocity in the drain pipe. Do not be come about drain pipes clogging?  

      Thank you for this observation, we obtained a velocity of 0.6 m/s at a water level of 130mm in that outlet and 1.03 m/s at a water level of 175mm in another outlet, which resulted in 0.0145 and 0.019 Manning number. Although the pipes in the AB5 sub-basin are PE corrugated, in the final point of the outlet, where we monitor the water level, its material is concrete.

      Regarding the drain pipes clogging we added in L211: covered by geotextile and gravels to avoid pipe clogging

Page 6 – 7, Table 3 (Sowing dates); I miss units

Thank you for the observation, we added month as unit in Table 3.

Most equations (formulas) do not content the units. E.g. formula (10)

WE = (I + R – D) x 100/ (I + R)

Units of WE, Water Efficiency??, in % ok; but units of I – total irrigation water (m3? litres? mm?

Units of R – total rainfall (mm??)

Units of D – total amount of drainage water (mm? litres/sec/ha or mm/day??

Thank you for the observation, as we had yearly irrigation data (volume used at the beginning and end of the irrigation period in each hydrant), we calculated the indexes above in mm/year, we added in L394: calculated during the study period with total amount of water (mm)

Etc., etc …..

Equations (11), (12); what is the reason of “P”? Where can I get it?

Thank you for the observation, that was a typographic mistake, as we first named rainfall (R) as precipitation (P) but we finally changed it. Now we changed both “P” to “R” in each equation.

Page 13, Figure 2

Q (m3/h) is water flow, total water flow or drainage water flow (page 12, line 439 – 444)?

Where is drawn the value of irrigation water? 

The range between individual days (day of the year – X axis) is very unobvious, e.g. 04-05 means the fourth of May?  

Figure 2 demonstrates “Day of the year”, but what year? One certain (concrete) year? Which one? Typical (representative) year from certain period? Please, can you specify this?

Thank you for the observations regarding Figure 3 (in the new version), we defined better the parameters and the year as the study period (2019-2020) and changed the format of the Day of the year in X axis.

The value of irrigation water, as we only have the use of it during the whole year in each farm, is not reflected in this figure, due to other changes in M&M, now is Figure 3.

Page 13, Chapter 3.3.1. HEC-HMS model (line 456 – 463) From what period are there the data mentioned in HEC-HMS model? Long term annual?

Thank you for this observation, we did miss to mention the period, as it is the study period. We added L506: during the study period (15 October 2019 to 15 October 2020)

Reviewer 2 Report

I have read the paper on drainage in a Mediterranean Semi Arid Basin, and I have a few core comments:

- Why don't you simply mention the basin in the title? In fact, it would make sense to have this current title if you were discussing different basins from the Mediterranean. Also, it would have been better to have them from both shores of the sea rather than just from one

- the introduction does a good job in contextualising the study; however, it does not tell us why this case study matters. Why is it relevant to understand the broader dynamics? in other words, can we generalise lessons relevant for all semi arid basins of the Mediterranean area? If yes, why and how?

- justify the research design and choice of case study

Author Response

First of all, thank you for the useful comments, we have done our best to clarify all the issues raised regarding the manuscript. Find attached the new version of the manuscript in which we added some information in order to clarify all the observation from you and your colleagues and improve the quality of the manuscript, and point-by-point responses to your comments, 

Why don't you simply mention the basin in the title? In fact, it would make sense to have this current title if you were discussing different basins from the Mediterranean. Also, it would have been better to have them from both shores of the sea rather than just from one

Thank you for this comment. We did not mention the basin in the title because the studied area is a small sub-basin (425 ha) and it has no common name, belonging to a bigger one, which is also being studied and these studies will be published soon, with mention directly to the basin in the title. We choose that title in order to highlight the fact that belongs to a mediterranean area (the Ebro basin), not only on basin but also on climate.

The introduction does a good job in contextualising the study; however, it does not tell us why this case study matters. Why is it relevant to understand the broader dynamics? in other words, can we generalise lessons relevant for all semi arid basins of the Mediterranean area? If yes, why and how?

Thank you for this comment, our hypotesis is that the leaching fractions (or overuse of water) in many irrigated sub-basins are higher than the necessaries. In general, when drainage water is quantified we do not know which fraction of it comes from rain and which one from irrigation. The use of the HEC-HMS model allowed us to have a first aproximation at this uncertainty, which we hope to adress in following studies with isotopic measurements. In our introduction we point in L60-64 the importance on drainage water and its monitoring. We think the information obtained is relevant because it can be transfered to the scientific community as well as to the involved stakeholders that the improvement of the irrigation management has still a margin. We also included a sencence in introducing the remote sensing concept in order to contextualise it more accurately.

Justify the research design and choice of case study

Thank you for this observation, the study area is a new irrigated area (20 years) with a well-defined and easily monitorable drainage network. Moreover, information is provided by the irrigation district fluently, as we point in the acknowledgements. We added in L216-217: These modern facilities enable monitoring and control of irrigation, rainfall and drainage water in the area. 

Reviewer 3 Report

The paper entitled “Understanding Drainage Dynamics and Irrigation Management in a Semi-arid Mediterranean Basin” by Altés et al could provide some information for the deep understanding on Drainage Dynamics and Irrigation Management. Topic is interesting and certainly in the scope of the Journal. In this paper, Water balance equation, the single-crop FAO-56 method , a two-source energy balance model (TSEB), HEC-HMS model , Sub-basin model and some other formulas were used to determine the amount of water lost to drainage in a semi-arid Mediterranean irrigated area. Although I really appreciate your work, the manuscript in this version is poorly written and data are poorly discussed. It needs major revision.

1. Major comments.

(1) The abstract does not match the content of the article, as well as the conclusion. In addition, The significance of the research is not reflected in the abstract.

L24-25:”These results indicate that the irrigation management and drainage dynamics in the studied basin are similar to other irrigated areas in the Ebro basin.” The purpose of this article is simply to verify this conclusion. Right?

(2) Please supplement the water table, Boundary of study area, and water supply network in the study area. Is groundwater involved in irrigation? How are the boundaries of the study area determined? How to consider the lateral exchange capacity at the boundary of the study area? Does the metered water supply network only cover the study area?

(3) The measurement of total drainage is not reliable. How to consider deep seepage?

(4) L320-322:The Hydros21 sensor measures water electrical conductivity corrected at standard temperature of 25(EC at dS/m), water temperature () and water depth (mm).Please add the water temperature and water depth parameters of Hydros2 sensor.

(5)L383-392: The number of measuring points is small, the measuring depth is shallow, So the representativeness is not strong.

(6) In this paper, HEC-HMS model is used to assess the fraction of drainage water that comes from rainfall. L464-465:”According to the model used, the contribution of rainfall to the total amount of drainage water in the sub-basin AB5 was estimated to be 41.4%.” Then, according to the water balance data measured at the sub-basin level, the 43% of the drainage water is rainfall drainage is obtained in L520. However, the reasons for the difference between the two outcomes are not analyzed, and no reason is given for the final result to adopt 43% . Please analyze and provide a detailed explanation.

2. Specific comments

(1) L164 : It is suggested that Table 1 be represented as a graph.

(2)L240-241: Add the case where R is equal to 75 mm.

(3)L334: “Drainage ratio (DR)” alone here are not standard.

(4) Table 1 and Table 3 are spread across pages. According to the text, L439:”Figure 3 shows the hourly evolution of water flow…” ,Figure 3 should be changed to Figure 2. L451:” Table 4. Average water level …”, Table 4 should be changed to Table 8. Please check the full text and correct it.

(5) L261-262:”where D is the total amount of drainage water. For pressurized irrigation systems, WE is normally considered to be greater than 75%.” Please provide a reference that WE is greater than 75%.

(6)L271-272:”This includes a first guess of canopy transpiration at the potential rates using an alpha coefficient initially set to 1.26.” Why is the initial alpha value set to 1.26? Please provide relevant references.

Author Response

First of all, thank you for the useful comments, we have done our best to clarify all the issues raised regarding the manuscript. Find attached the new version of the manuscript in which we added some information in order to clarify all the observation from you and your colleagues and improve the quality of the manuscript, and point-by-point responses to your comments.

  1. Major comments.

(1) The abstract does not match the content of the article, as well as the conclusion. In addition, The significance of the research is not reflected in the abstract.

L24-25:”These results indicate that the irrigation management and drainage dynamics in the studied basin are similar to other irrigated areas in the Ebro basin.” The purpose of this article is simply to verify this conclusion. Right?

Thank you for this important observation, as you point, one of the purposes of the article is to verify that the management and drainage dynamics in the area are similar to other irrigated basins in the Ebro basin. However, other points are addressed as the fraction of rainwater and irrigation water that reaches the outlet, with the possibility to discriminate between them and assess the irrigation efficiency considering the irrigation drainage water losses. That is the main point in the abstract, in which we highlight the irrigation efficiency and the total water lost through drainage (L19-23). In the sentence L24-25 we state that being similar to other basins may indicate that the actions to be developed in the basin to improve the irrigation management may be similar to others in others sub-basin in the area.

After your observation, we realized that the LR and LF were not compared during the discussion (added L597-601) and either the conclusions (L648-652), we added information regarding to that and also added a statement in the abstract about it (L20-21)

(2) Please supplement the water table, Boundary of study area, and water supply network in the study area. Is groundwater involved in irrigation? How are the boundaries of the study area determined? How to consider the lateral exchange capacity at the boundary of the study area? Does the metered water supply network only cover the study area?

Thank you for your interesting questions, we explained on detail the geology, hydrogeology and soil hydraulic proprieties of the area and the studied sub-basin in order to provide enough information to the reader regarding the water table and other hydraulic parameters.

Regarding irrigation, as we indicate in L187 in the revised version: Water used in the irrigation district is pumped 32 m from the Noguera Ribagorçana River a highly regulated river with four dams, belonging to the Ebro River basin

On the other hand, the metered water supply network covers all the irrigation district, as we point in L201-205 of the revised version: All the irrigation systems in the area are pressurized, with sprinkler and drip irrigation systems used for herbaceous and woody crops, respectively. Water is supplied to the farm fields by delivery points (hydrants). The total number of hydrants is 1,351, making an average of 5.07 ha/hydrant. These delivery points also facilitate the supply of water to pig, cattle and poultry farms. However, we studied only the AB5 sub-basin because it was well delineated on the GIS and the influence of the aquifers from the two boundary rivers is null.

Regarding the boundary of study area, we rely of the delineation of the sub-basin using the algorithm developed by Wang & Liu (2006), which works with DEM and is one of the main tools for GIS hydrologic analysis

(3) The measurement of total drainage is not reliable. How to consider deep seepage?

Thank you for the observation, in the revised version we added the detailed hydrogeological and soil hydraulic proprieties of the study area L116-171, in which we indicate that the deep seepage and capillary rise is considered as null, due to the low influence from the aquifers from the Segre and Noguera Ribagorçana rivers.

(4) L320-322:”The Hydros21 sensor measures water electrical conductivity corrected at standard temperature of 25℃(EC at dS/m), water temperature (℃) and water depth (mm).”Please add the water temperature and water depth parameters of Hydros2 sensor.

Thank you for the comment, but I think I do not understand it. Water depth values are shown in Table 8 L502 as “y”, which is latter expressed as drainage discharge as “Qd” (the “d” subindex was added after the revisions). We did not present Temperature and EC because it was not the aim of this study.

(5)L383-392: The number of measuring points is small, the measuring depth is shallow, So the representativeness is not strong.

Thank you for this observation, we agree that the number of measurements is small, but we tried to install them in two farms (20 ha and 37 ha) representative from the study area (425 ha), also the depth is shallow due to the fact that mainly irrigation is sprinkler, with a high-frequency (irrigation every day during the maximum requirement stages). Is true that the representativeness may be weak so one option would be assume that there is no variation on soil water content, as the study is developed during one hydrological year. Barros et al. (2011) indicate in a similar study that the season average change in soil water content is a minor value compared to the rest of terms in the equation of water balance. In the revised version we retired the ∆SWC and assumed it as 0 during the period.

(6) In this paper, HEC-HMS model is used to assess the fraction of drainage water that comes from rainfall. L464-465:”According to the model used, the contribution of rainfall to the total amount of drainage water in the sub-basin AB5 was estimated to be 41.4%.” Then, according to the water balance data measured at the sub-basin level, the 43% of the drainage water is rainfall drainage is obtained in L520. However, the reasons for the difference between the two outcomes are not analyzed, and no reason is given for the final result to adopt 43% . Please analyze and provide a detailed explanation.

Thank you for the observation, we had two mathematical mistakes involved, the first one in 3.3.1.where the estimated contribution of rainfall to drainage was 42.2%. The other one was due to that in 3.6 table 11 we rounded the irrigation drainage contribution to 57%. To avoid further mistakes we changed 3.6 not presenting it as a table. In the new version data is corrected and the real result is 42.4%. With that we also detected some mistake that made differ slightly the final results.

  1. Specific comments

(1) L164 : It is suggested that Table 1 be represented as a graph.

Thank you for this observation, we had also though about that, we have changed it to a graph (bar-points).

(2)L240-241: Add the case where R is equal to 75 mm.

We changed ER = 0.6 · R – 10 when R ≤ 75 mm

(3)L334: “Drainage ratio (DR)” alone here are not standard.

Thank you for the observation, Drainage ratio, rainwater drainage ratio and irrigation drainage ratio were listed with bullets as a list according to the “Water” edition requirements

(4) Table 1 and Table 3 are spread across pages. According to the text, L439:”Figure 3 shows the hourly evolution of water flow…” ,Figure 3 should be changed to Figure 2. L451:” Table 4. Average water level …”, Table 4 should be changed to Table 8. Please check the full text and correct it.

In the revised version we tried to be more careful in the format, we hope we don’t have this mistakes now, thanks for the observation.

(5) L261-262:”where D is the total amount of drainage water. For pressurized irrigation systems, WE is normally considered to be greater than 75%.” Please provide a reference that WE is greater than 75%.

Thank you for this observation, in the case of Water efficiency with this calculation, we wanted to point that our objective would be to reach a 75% of value, with no reference on it due to the calculations on drainage as we did are not that common. We deleted the sentence in which we state that may be greater than 75%.

(6)L271-272:”This includes a first guess of canopy transpiration at the potential rates using an alpha coefficient initially set to 1.26.” Why is the initial alpha value set to 1.26? Please provide relevant references.

Thank you for the observation, the Priestley-Taylor approach uses an alpha coefficient of 1.26. The Priestaly-taylor approach has been widely calibrated in multiple crops and environments, all of them explained in Priestley and Taylor (1972). Further, Kustas and Norman (1999) also used this alpha coefficient according to Priestley-Taylor (1972). In order to clarify that, we added both references in L332.

Reviewer 4 Report

1.      Why the study area is not expressed in longitude and latitude.

2.      Why is there no introduction to remote sensing data?

3.      Line238-241 It is necessary to know the quantitative results ‘which has agood accuracy according to Kumar et al.’

4.      Line287-289 It is necessary to supplement the corresponding LAI and VCF. ‘Among different biophysical parameters, the leaf area index (LAI) and the fraction of vegetation cover (fc) were used as the inputs of the two-source energy balance (TSEB) model.’

5.      Need to improve the definition of fig3.

6.      Why is the content of Table 11 expressed in a table?

7.      The conclusion lacks quantitative description support

Author Response

First of all, thank you for the useful comments, we have done our best to clarify all the issues raised regarding the manuscript. Find attached the new version of the manuscript in which we added some information in order to clarify all the observation from you and your colleagues and improve the quality of the manuscript, and point-by-point responses to your comments.

  1. Why the study area is not expressed in longitude and latitude.

Thank you for this observation, we decided to use the UTM coordinates because we were working with GIS data, for basin definition, soil and geological characteristics, which is georeferenced mainly with UTM. If necessary, we could also add the longitude and latitude.

  1. Why is there no introduction to remote sensing data?

Thank you for the comment, indeed we do not introduce remote sensing, as we talk directly on the role of it in irrigation management, after your observation, we included some comments on it in L82-86: One of the tools that is being used largely for studying the environment and landscape is remote sensing, which is the collection and interpretation of information about an object, area or event without being in physical contact with it [21]. The term “remote sensing” was first coined in the early 1960s to describe any means of observing the Earth from afar, particularly as applied to aerial photography, the main sensor used at that time [22].

  1. Line238-241 It is necessary to know the quantitative results ‘which has agood accuracy according to Kumar et al.’

Thank you for the observation, in the article of Kumar et al. they review the methods in Effective rainfall calculations with no support with quantitative data, we also added the comment: which uses parameters of climate, crop and soil data, is suitable for all kind of situations and has a good accuracy according to Kumar et al. in L296-298 in the revised paper.

  1. Line287-289 It is necessary to supplement the corresponding LAI and VCF. ‘Among different biophysical parameters, the leaf area index (LAI) and the fraction of vegetation cover (fc) were used as the inputs of the two-source energy balance (TSEB) model.’

Thank you for this comment, In the sentence previous to this one, it is already explained that biophyisical parameters of the vegetation (such as LAI and FVC) were obtained from the biophysical processor SNAP, which applies the PROSAIL radiative transfer model and machine learning approaches in Sentinel-2 data. References are also included.

  1. Need to improve the definition of fig3.

Thank you for this observation, in the new version we tried to define better (X axis) the time and the definition of the parameters.

  1. Why is the content of Table 11 expressed in a table?

Thank you for this observation, we changed it and presented as regular text, and also changed the dimensions to m3 to avoid lost in decimal data

  1. The conclusion lacks quantitative description support

Thank you for this important comment, after it we revised the manuscript and realized we did not address the differences between the DR and the LR, which we talk about in the introduction. After that we tried to improve the discussion (L597-601) and the conclusion (L648-652) and abstract (L20)

Round 2

Reviewer 1 Report

The manuscript was obviously improved and is very suitable to “Water MDPI – Journal”. Some of notes was ignored, I am still missing following:

 Drainage

-        I miss the drain spacing of ditches (distance between individual ditches /open drains/), how are their values and where the design of drainage system comes from?

-        I miss the drain spacing of subsurface pipes (distance between individual pipes /subsurface pipe drainage system/), how are their values and where the design of subsurface drainage system comes from?

-        You mentioned Chezy’s Equation with Manning formula (page 18, Appendix A) to calculate velocity of the drain pipes. Can you specify the results? Means concrete value of velocity in the drain pipe.

Page 8, Table 3 (Sowing date); I still miss units

 Most equations (formulas) do not content the units. E.g. formula (10), page 9,

 WE = (I + R – D) x 100/ (I + R)

Units of WE, Water Efficiency??, in % ok; but units of I – total irrigation water (m3? litres? mm?

Units of R – total rainfall (mm??), units?

Units of D – total amount of drainage water, what units?? (mm? litres/sec/ha or mm/day??)

Etc., etc …..

Equations (11), (12), page 10; what is the units of DR, respectively RWDR?

Author Response

The manuscript was obviously improved and is very suitable to “Water MDPI – Journal”. Some of notes was ignored, I am still missing following:

 Drainage

-        I miss the drain spacing of ditches (distance between individual ditches /open drains/), how are their values and where the design of drainage system comes from

-        I miss the drain spacing of subsurface pipes (distance between individual pipes /subsurface pipe drainage system/), how are their values and where the design of subsurface drainage system comes from?

Thank you again for these observations, as pointed in the last answer, the public regional infrastructure company was in charge on the design of the main drainages and ditches. After checking the documents on the actions they developed, we can affirm that the values of designing and dimensions are following the Norma 5.2–IC de la Instrucción de Carreteras. Drenaje superficial (carreteros.org)  and Maximas lluvias diarias (mitma.gob.es) from the Spanish authorities. In the case of the studied sub-basin, the main subsurface pipe has a 400 mm DN with ID (internal diameter) of 348 mm, and reaches the concrete outlet with an extended diameter of 370 mm.

      Drainage tiles have been partially installed in some agricultural fields to lower the water table, thus improving growing conditions. They are installed at a depth of 75 to 100 cm and are connected to the general drainage network. However, we do not know in detail which fields have installed such drains.

-        You mentioned Chezy’s Equation with Manning formula (page 18, Appendix A) to calculate velocity of the drain pipes. Can you specify the results? Means concrete value of velocity in the drain pipe.

      Thank you again for the observation, we added L734-L739: In these measurements we obtained velocities of 0.6 m/s with a water level of 130 mm in the studied outlet and 1.03 m/s with a water level of 175 mm in another outlet, which resulted in 0.0145 and 0.019 Manning number (n), respectively. Although the pipes in the AB5 sub-basin are mainly polyethylene, in the final point of the outlet, where water is monitored, the material is concrete, being realistic the Manning number obtained.

Page 8, Table 3 (Sowing date); I still miss units

Thank you again for the observation. As each farmer decided which day to sow, we made an approximation in the month, as we cannot know exactly the day. In the new version we changed date for time, to avoid misunderstandings. The lengths of each stage are in days, as indicated in the table description.

 Most equations (formulas) do not content the units. E.g. formula (10), page 9,

 WE = (I + R – D) x 100/ (I + R)

Units of WE, Water Efficiency??, in % ok; but units of I – total irrigation water (m3? litres? mm?

Units of R – total rainfall (mm??), units?

Units of D – total amount of drainage water, what units?? (mm? litres/sec/ha or mm/day??)

Etc., etc …..

Equations (11), (12), page 10; what is the units of DR, respectively RWDR?

Thank you again for these important observations, after checking it again, we think we have now defined all indexes and abbreviations:

L300 (ETc in mm),

L314 (NIR in mm),

L316 ER (in mm),

 L322 “where R is the accumulated monthly rainfall in mm”,

L323 (GIR in mm).

L331: Where I is the amount of water applied as irrigation (mm)

L343: where D is the total amount of drainage water (in mm)

L423: where RWD is the amount of rainwater drained in mm

L428: where ID is the amount of irrigation water drained in mm

After that, we also added L429: Therefore, D is the sum of RWD and ID.

Also, in order to clarify the results, we improved the explanation in L601-L609: On the other hand, the total amount of drained water (D) corresponds to 862,142 m3, being DR 18%. Moreover, ID corresponds to 496,217 m3, being IDR 19.3% and RWD to 365,925 m3 being RWDR 16.2%.

Therefore, we could state that the 57.6% of the drainage water is ID and the 42.4% remaining is RWD.

The leaching requirements (LR) in the study area for an ECw of 0.37 dS/m and ECe of 2dS/m is 3.8%, a significantly lower value than the obtained IDR.

On another hand, we changed slightly for better understanding the conclusions, L708-L713: With the present methodology, we estimated a loss of water through drainage (DR) of 18%. The present study permitted to estimate what percentage of the total drainage water comes from rainwater (42.4%) and what percentage comes from irrigation (57.6%). The only way to reduce drainage losses is through improvement of irrigation management. The available margin of improvement is between 19.3% of the present IDR and the 3.8% estimated with the theoretical LR model.

Finally, to clarify better the conclusion, we changed it in the abstract L24-25: The available margin of improvement is between 19.3% of the present irrigation drainage ratio and the 3.8% estimated with the leaching requirement model

Reviewer 2 Report

My previous comments were not accounted for and not incorporated, so i would suggest revisions. The title should be specific, now it is very general "a basin". This is just one example. 

Author Response

My previous comments were not accounted for and not incorporated, so i would suggest revisions. The title should be specific, now it is very general "a basin". This is just one example. 

Thank you again for your observations, we keep in our idea on maintaining the present title of the manuscript, as we think the basin is not that important to be cited in the title (it has 425 ha). After revising the manuscript, we saw some other problems and we revised several sections so that they were tighter and more to the point we want to address.

Previously we had the main conclusion as L26-L27:

These results indicate that the irrigation management and drainage dynamics in the studied area are similar to other irrigated areas in the Ebro basin. This sentence could lead to think that the basin studied reflects the behavior of other basins of the Mediterranean, and we better do not dare to say this statement.

In order to avoid misunderstanding, we changed the main conclusion reflected in the abstract L24-25: The available margin of improvement is between 19.3% of the present irrigation drainage ratio and the 3.8% estimated with the leaching requirement model.

Also we improved the presentation of the results in order to clarify them, L601-609:

On the other hand, the total amount of drained water (D) corresponds to 862,142 m3, being DR 18%. Moreover, ID corresponds to 496,217 m3, being IDR 19.3% and RWD to 365,925 m3 being RWDR 16.2%.

Therefore, we could state that the 57.6% of the drainage water is ID and the 42.4% remaining is RWD.

The leaching requirements (LR) in the study area for an ECw of 0.37 dS/m and ECe of 2dS/m is 3.8%, a significantly lower value than the obtained IDR.

Moreover, we addressed the discussion in the direction on comparing the present irrigation drainage with the theoretical one (leaching requirement), changing some sentences in L656-L660: If we compare the IDR with the LR, we can state that the leaching requirements were highly accomplished, with a surplus of almost five time more water used for leaching than the theoretical required ones. This means that the improvement of irrigation management ant the reduction of water loss may go through the direction on reducing the actual irrigation drainage ratio (IDR).

Finally, we changed also the main conclusion of the study by insisting in the fact that the improvement of irrigation will occur by reducing the present irrigation drainage ratio L708-713: With the present methodology, we estimated a loss of water through drainage (DR) of 18%. The present study permitted to estimate what percentage of the total drainage water comes from rainwater (42.4%) and what percentage comes from irrigation (57.6%). The only way to reduce drainage losses is through improvement of irrigation management. The available margin of improvement is between 19.3% of the present IDR and the 3.8% estimated with the theoretical LR model.

Reviewer 3 Report

Accept !

Author Response

Thank you for your previous comments that enriched the manuscript, and also for the acceptance in the present review. After revising again for the present responses, we saw some other details and revised the results and conclusions so that they were tighter and more to the point. 

Previously we had the main conclusion as L26-L27:

These results indicate that the irrigation management and drainage dynamics in the studied area are similar to other irrigated areas in the Ebro basin. This sentence could lead to think that the basin studied reflects the behavior of other basins of the Mediterranean, and we better do not dare to say this statement.

In order to avoid misunderstanding, we changed the main conclusion reflected in the abstract L24-25: The available margin of improvement is between 19.3% of the present irrigation drainage ratio and the 3.8% estimated with the leaching requirement model.

Also we improved the presentation of the results in order to clarify them, L601-609:

On the other hand, the total amount of drained water (D) corresponds to 862,142 m3, being DR 18%. Moreover, ID corresponds to 496,217 m3, being IDR 19.3% and RWD to 365,925 m3 being RWDR 16.2%.

Therefore, we could state that the 57.6% of the drainage water is ID and the 42.4% remaining is RWD.

The leaching requirements (LR) in the study area for an ECw of 0.37 dS/m and ECe of 2dS/m is 3.8%, a significantly lower value than the obtained IDR.

Moreover, we addressed the discussion in the direction on comparing the actual irrigation drainage with the theoretical one (leaching requirement), changing some sentences in L656-L660: If we compare the IDR with the LR, we can state that the leaching requirements were highly accomplished, with a surplus of almost five time more water used for leaching than the theoretical required ones. This means that the improvement of irrigation management ant the reduction of water loss may go through the direction on reducing the actual irrigation drainage ratio (IDR).

Finally, we changed also the main conclusion of the study by insisting in the fact that the improvement of irrigation will occur by reducing the present irrigation drainage ratio L708-713: With the present methodology, we estimated a loss of water through drainage (DR) of 18%. The present study permitted to estimate what percentage of the total drainage water comes from rainwater (42.4%) and what percentage comes from irrigation (57.6%). The only way to reduce drainage losses is through improvement of irrigation management. The available margin of improvement is between 19.3% of the present IDR and the 3.8% estimated with the theoretical LR model.

Reviewer 4 Report

The paper is written in good quality.

Author Response

Thank you for your previous comments that enriched the manuscript, and also for the acceptance in the present review. After revising again for the present responses, we saw some other problems and revised the results and conclusions so that they were tighter and more to the point. 

Previously we had the main conclusion as L26-L27:

These results indicate that the irrigation management and drainage dynamics in the studied area are similar to other irrigated areas in the Ebro basin. This sentence could lead to think that the basin studied reflects the behavior of other basins of the Mediterranean, and we better do not dare to say this statement.

In order to avoid misunderstanding, we changed the main conclusion reflected in the abstract L24-25: The available margin of improvement is between 19.3% of the present irrigation drainage ratio and the 3.8% estimated with the leaching requirement model.

Also we improved the presentation of the results in order to clarify them, L601-609:

On the other hand, the total amount of drained water (D) corresponds to 862,142 m3, being DR 18%. Moreover, ID corresponds to 496,217 m3, being IDR 19.3% and RWD to 365,925 m3 being RWDR 16.2%.

Therefore, we could state that the 57.6% of the drainage water is ID and the 42.4% remaining is RWD.

The leaching requirements (LR) in the study area for an ECw of 0.37 dS/m and ECe of 2dS/m is 3.8%, a significantly lower value than the obtained IDR.

Moreover, we addressed the discussion in the direction on comparing the actual irrigation drainage with the theoretical one (leaching requirement), changing some sentences in L656-L660: If we compare the IDR with the LR, we can state that the leaching requirements were highly accomplished, with a surplus of almost five time more water used for leaching than the theoretical required ones. This means that the improvement of irrigation management ant the reduction of water loss may go through the direction on reducing the actual irrigation drainage ratio (IDR).

Finally, we changed also the main conclusion of the study by insisting in the fact that the improvement of irrigation will occur by reducing the present irrigation drainage ratio L708-713: With the present methodology, we estimated a loss of water through drainage (DR) of 18%. The present study permitted to estimate what percentage of the total drainage water comes from rainwater (42.4%) and what percentage comes from irrigation (57.6%). The only way to reduce drainage losses is through improvement of irrigation management. The available margin of improvement is between 19.3% of the present IDR and the 3.8% estimated with the theoretical LR model.